# Structural basis for energy transfer in a huge diatom PSI-FCPI supercomplex

Caizhe Xu 1,2,8, Xiong Pi3,8, Yawen Huang3,8, Guangye Han1, Xiaobo Chen 1,4, Xiaochun Qin5, Guoqiang Huang3, Songhao Zhao1,2, Yanyan Yang1, Tingyun Kuang 1, Wenda Wang 1✉, Sen-Fang Sui3,6✉ & Jian-Ren Shen 1,7✉

Diatom is an important group of marine algae and contributes to around 20% of the global photosynthetic carbon fixation. Photosystem I (PSI) of diatoms is associated with a large number of fucoxanthin-chlorophyll *a/c* proteins (FCPIs). We report the structure of PSI-FCPI from a diatom *Chaetoceros gracili*s at 2.38 Å resolution by single-particle cryo-electron microscopy. PSI-FCPI is a monomeric supercomplex consisting of 12 core and 24 antenna subunits (FCPIs), and 326 chlorophylls *a*, 34 chlorophylls *c*, 102 fucoxanthins, 35 diadinoxanthins, 18 *β*-carotenes and some electron transfer cofactors. Two subunits designated PsaR and PsaS were found in the core, whereas several subunits were lost. The large number of pigments constitute a unique and huge network ensuring efficient energy harvesting, transfer and dissipation. These results provide a firm structural basis for unraveling the mechanisms of light-energy harvesting, transfer and quenching in the diatom PSI-FCPI, and also important clues to evolutionary changes of PSI-LHCI.

[1] Photosynthesis Research Center, Key Laboratory of Photobiology, Institute of Botany, The Chinese Academy of Sciences, Beijing 100093, China. [2] University of Chinese Academy of Sciences, Yuquan Road, Shijingshan District, 100049 Beijing, China. [3] State Key Laboratory of Membrane Biology, Beijing Advanced Innovation Center for Structural Biology & Frontier Research Center for Biological Structure, School of Life Sciences, Tsinghua University, Beijing 100084, China. [4] School of Biological Science and Engineering, Hebei University of Science and Technology, Shijiazhuang 050018, China. [5] School of Biological Science and Technology, University of Jinan, Jinan 250022, China. [6] Department of Biology, Southen University of Science and Technology, Shenzhen, Guangdong 518055, China. [7] Research Institute for Interdisciplinary Science, and Graduate School of Natural Science and Technology, Okayama University, Okayama 700-8530, Japan. [8] These authors contributed equally: Caizhe Xu, Xiong Pi, Yawen Huang. ✉email: wdwang@ibcas.ac.cn; suisf@mail.tsinghua.edu.cn; jrshen@ibcas.ac.cn

Oxygenic photosynthesis provides the source of energy and oxygen indispensable for sustaining almost all life forms on the earth. Two photosystems (PSs), PSI and PSII, are responsible for the primary photochemical reactions of photosynthesis[1]. Each PS is associated with variable numbers and compositions of light-harvesting pigment-protein complexes (LHCs) that serve as light-harvesting antennas. Among the two PSs, PSI drives electron transfer derived from water oxidation by PSII to ferredoxin, leading to the generation of reducing power needed for reducing $CO_2$ into sugars[1]. The core components of PSI are largely conserved during evolution[2–4], although some core subunits are recruited or lost, and the oligomerization state of the PSI core is also changed among prokaryotic and eukaryotic organisms[2–13]. The most remarkable changes are found in LHC proteins and pigments they bind. Cyanobacterial PSI has no transmembrane LHC and sometimes is associated with hydrophilic phycobilisomes as its antenna[14], whereas PSI of eukaryotic organisms has membrane-spanning LHCs as their antenna[8–13]. The light-harvesting complex I (LHCI) in eukaryotic organisms differ remarkably in their sequences, numbers bound to each PSI core, and the pigments they associate. Red algal PSI has either three or five red algal-type LHC (LHCR) antenna subunits in which chlorophyll (Chl) *a* and zeaxanthins are main light-harvesting pigments[8,15,16]. LHCI of both green algae and higher plants in the green lineage binds Chl *a/b* and carotenoids (Cars), but they differ in the number of the LHCI subunits bound to each PSI core remarkably: green algal PSI contains up to ten LHCI subunits[11–13], whereas higher plant PSI normally contains four LHCI subunits[9,10]. In the red lineage, many eukaryotic algae arise from secondary endosymbiosis and have unique LHCs, which bind Chl *a/c* and fucoxanthins (Fx), and are designated FCP (fucoxanthin Chl *a/c*-binding protein)[17–23]. Chl *a/c* and Fx has rather strong absorptions in the region of 400–550 nm, enabling these organisms survival efficiently under aquatic environments where red light is diminished and the light in the region of 400–550 nm is more available[24–26].

Diatoms are a main group of the red lineage and play important roles in oxygen production, $CO_2$ fixation, and biogeochemical cycle of carbon and silica, which account for about 20% of the global primary productivity[27]. Both PSI and PSII of diatoms bind FCPs as their light-harvesting antennas[22,23,28,29], conferring a brown color to the algae different from the green color seen in the green lineage organisms. The structure of diatom PSII-FCPII has been solved by cryo-electron microscopy (cryo-EM) recently[22,23], revealing a unique FCPII organization around the PSII core and detailed binding sites of Chl *a/c* and Fxs. On the other hand, diatom PSI binds more than 20 FCPI antenna subunits[19,29,30], making it the largest antenna system found in the eukaryotic photosynthetic organisms and supporting the efficient survival of diatoms in the aquatic environment[31–34]. Here we solved the structure of PSI-FCPI supercomplex from a diatom *Chaetoceros gracilis* at 2.38 Å resolution by cryo-EM, which showed many unique features of the diatom PSI core and the binding of 24 FCPI subunits surrounding the core, providing a solid structural basis for revealing the energy harvesting, transfer, and dissipation mechanisms in the diatom PSI-FCPI supercomplex.

## Results and discussion

**Overall structure**. PSI-FCPI supercomplex was purified from a centric diatom *C. gracilis* (Supplementary Fig. 1a) and its peptide composition, spectroscopic properties, and pigment composition are shown in Supplementary Fig. 1b–e. Structure of the PSI-FCPI supercomplex was solved by cryo-EM at an overall resolution of 2.38 Å and a local resolution of 2.2 Å for the PSI core based on

the "gold standard" of Fourier shell correlation (FSC) = 0.143[35] (Table 1 and Supplementary Fig. 2). The overall structure of the *C. gracilis* PSI-FCPI is an asymmetric heart-shaped monomer when viewed from the stromal side (Fig. 1a), in which the PSI core is composed of 12 subunits, including PsaA/B/C/D/E/F/I/J/ L/M subunits and 2 previously unidentified subunits PsaR and PsaS (Fig. 1 and Supplementary Table 1). This monomeric PSI core is surrounded by 24 FCPI antennas (Fig. 1 and Supplementary Fig. 3a), which are distributed in an innermost layer, a semi-ring in the middle layer, and three FCPIs in the outermost layer (Fig. 1a, b, d).

In addition to the protein subunits, we identified 326 Chls *a*, 34 Chls *c*, 102 Fx, 35 Ddx, 18 β-carotene (Bcr), 2 phylloquinones, 3 $Fe_4S_4$ clusters, 69 lipids, a number of detergents, and water molecules (Table 1 and Supplementary Table 1), giving rise to a supercomplex having 36 subunits and more than 500 cofactors with an overall molecular weight of 1.1 MDa. This diatom PSI-FCPI is the largest monomeric PSI supercomplex with the largest number of antennas attached among all of the photosynthetic organisms known so far.

**Table 1 Cryo-EM data collection, refinement, and validation statistics.**

|  | PSI-FCPI (EMDB-30012), (PDB 6LY5) |
|---|---|
| **Data collection and processing** | |
| Magnification | ×105,000 |
| Voltage (kV) | 300 |
| Electron exposure (e⁻/Å²) | 50 |
| Defocus range (μm) | −1.5 ~ −2.5 |
| Pixel size (Å) | 1.091 |
| Symmetry imposed | C1 |
| Number of initial particles | 891,804 |
| Number of final particles | 164,480 |
| Map resolution (Å) | 2.38 |
| FSC threshold | 0.143 |
| **Refinement** | |
| Initial model used (PDB code) | 5ZGB |
| Model resolution (Å) | 2.38 |
| FSC threshold | 0.143 |
| Map sharpening *B* factor (Å²) | −41.16 |
| **Model composition** | |
| Non-hydrogen atoms | 814,10 |
| Protein residues | 6645 |
| Ligands | |
| Chl *a* | 326 |
| Chl *c* | 34 |
| Fx | 102 |
| Bcr | 18 |
| Ddx | 35 |
| Lipids | 69 |
| Water | 153 |
| ***B* factors (Å²)** | |
| Protein | 56.94 |
| Ligand | 60.73 |
| **Root mean square deviations** | |
| Bond lengths (Å) | 0.012 |
| Bond angles (°) | 1.592 |
| **Validation** | |
| MolProbity score | 2.08 |
| Clashscore | 10.57 |
| Poor rotamers (%) | 0.12 |
| **Ramachandran plot** | |
| Favored (%) | 90.33 |
| Allowed (%) | 9.22 |
| Disallowed (%) | 0.46 |

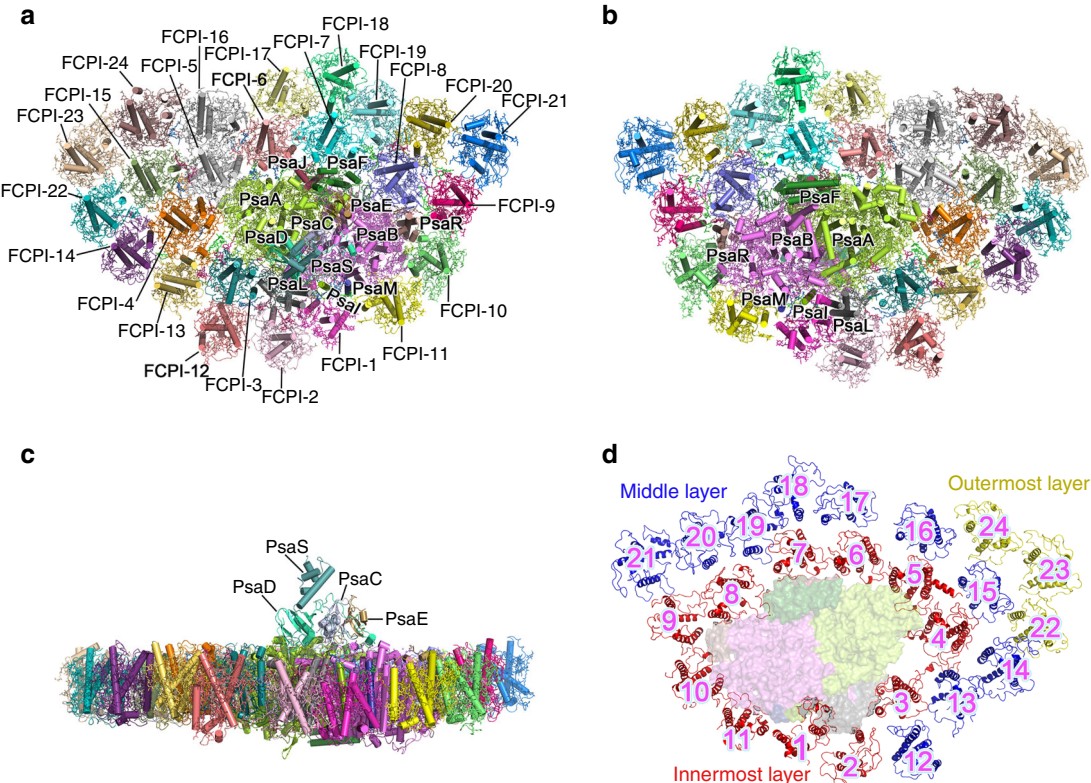

**Fig. 1 Overall structure of the PSI-FCPI supercomplex. a** Top view of the PSI-FCPI supercomplex from stromal side. **b** Bottom view from stromal side. **c** Side view of the PSI-FCPI supercomplex. **d** Three FCPIs layers of PSI-FCPI supercomplex. Red cartoon, innermost layer; blue cartoon, middle layer; yellow cartoon, outermost layer. Numbers in **d** indicate 24 FCPIs around the PSI core. Color codes for the PSI core subunits in **a**, **b**, and **c**: limon, PsaA; violet, PsaB; blue–white, PsaC; green–cyan, PsaD; sand, PsaE; forest, PasF; dirty violet, PsaH; splitpea, PsaI; raspberry, PsaJ; grey50, PsaL; deep blue, PsaM; dirty violet, PsaR; and light teal, PsaS. FCPI subunits are labeled in **a**, **b**, and **c**.

Among the 12 PSI core subunits, nine (PsaA/B/C/D/E/F/I/J/L) are conserved from cyanobacteria to higher plants (Supplementary Fig. 4)[3–13]; they are responsible for charge separation and electron transfer reactions and/or stabilization of the core structure. A small membrane-spanning subunit PsaM found in cyanobacteria and red algae, but absent in green algae and higher plants, is found in the diatom PSI. However, the diatom PSI lacks PsaG/H/K/N/O/X subunits, among which PsaX is present only in cyanobacterial PSI, whereas PsaG/H/N are found only in green algae and higher plants. Surprisingly, PsaK is present in all oxyphototrophs and PsaO is present in all eukaryotic oxyphototrophs, but these two subunits are absent in the diatom PSI core (Fig. 1 and Supplementary Fig. 4). The corresponding genes of these subunits are also absent in the diatom genomes[36–38]. These features may reflect the unique position of diatoms arising from secondary endosymbiosis of red algae.

Two previously unidentified subunits designated PsaR and PsaS were found in the diatom PSI core (Figs. 1 and 2). PsaR (chain h in the PDB file) is a transmembrane subunit with two transmembrane helices and binds one Chl *a* and one Fx. Its position is close to, but not overlapped with, that of PsaG in the green lineage organisms (Fig. 2c)[9–13]. Moreover, both N- and C-terminals of PsaR is located at the lumenal side, which is opposite to the N- and C-terminals of PsaG. PsaR has 89 residues and homologous sequences are found in other 3 diatom species (Supplementary Fig. 5c), but its homology to PsaG is very low (Supplementary Fig. 5d). PsaR binds to the PSI core through its interactions with PsaB by a hydrogen bond and hydrophobic interactions (Fig. 3a), and a salt bridge was formed between PsaR-Glu7 and FCPI-8-R151 at the stromal surface, which facilitates

binding of the peripheral FCPI antennas (Fig. 3c). These suggest a role of PsaR in mediating binding of FCPI and energy transfer from FCPI to the PSI core, resembling the role of PsaG in the green lineage organisms.

The second subunit is an extrinsic subunit PsaS (chain g in the PDB file) (Figs. 1c and 2d–f). PsaS was built as polyalanines, as its sequences could not be assigned in the structure due to its peripheral location. No ligands were found in PsaS (Fig. 2d) and it has a long C-terminal loop that anchors it to the surface of PsaB/C/D/L at the stromal surface (Fig. 2e). There are three short helices in the N-terminal region of PsaS, which are located at the top of PsaD and may protect PsaD from being exposed to the stromal solution (Fig. 2e, f). PsaS may have some protective effect to make PsaC/D/E stable; however, its relationship with the electron transfer out of PSI is still unknown.

The number of Chls *a* bound to the diatom PSI core is 94, which is 2 fewer than those in cyanobacteria and higher plant PSIs due to the lack of PsaG, PsaH, PsaK, and PsaO subunits[5,9,10]. Interestingly, two Ddx and two Fx were identified in the diatom PSI core, which are bound to PsaA/B/R based on the high-quality cryo-EM map (Supplementary Table 1). These Cars substitute for Bcrs in the PSI core of other organisms, which is probably related with the unique photoprotection property of the diatom PSI core.

**Location and structures of FCPI subunits.** A total of 24 FCPI subunits encircle the diatom PSI core and form 3 layers (Fig. 1a, b, d). This is eight more than the recently reported PSI-LHCI supercomplex structure (FCPI-1/2/12/17/18/19/20/21)[39]. The reason for this discrepancy may be due to the detachment of

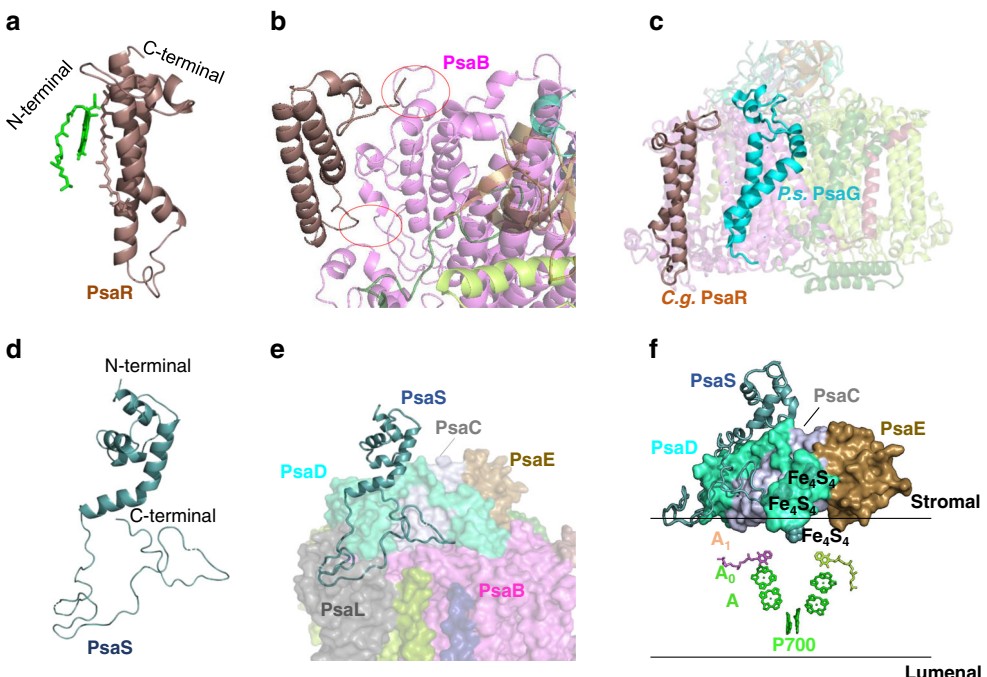

**Fig. 2 Previously unidentified subunits PsaS and PsaR in the diatom PSI core. a** Structure of PsaR. **b** Location of PsaR in the PSI core and its interaction with PsaB. **c** Comparison of the location of PsaR in *C. gracilis* PSI and PsaG in *P. sativum* PSI[9]. **d** Structure of PsaS. **e** Position of PsaS in the PSI core and the relationship between PsaS C-terminal and surrounding subunits. **f** Relative position of PsaS against the electron transfer chain.

some FCPI during isolation of PSI-LHCI or growth of the *C. gracilis* cells under a higher light intensity, in the latter case[40]. The 24 FCPI subunits are located in 3 layers based on their proximity to the PSI. The innermost layer of FCPI forms a closed ring consisting of 11 FCPI subunits (FCPI-1 to FCPI-11), the middle layer is a semi-cycled ring consisting of 10 FCPI subunits (FCPI-12 to FCPI-21), and an additional, outermost layer consists of 3 FCPI subunits (FCPI-22 to FCPI-24) and is located at the PsaA side with the largest distance of around 150 Å to the reaction center. Among these FCPIs, five (FCPI-1/5/6/7/11) share similar positions with five Lhcrs of red algal PSI-LHCI, but all FCPIs have shifted or are in completely different positions in comparison with the LHCI subunits in green algal and higher plant PSI-LHCIs (Supplementary Fig. 4).

The structures of 19 FCPI subunits were built from sequences of *C. gracilis* obtained from transcriptome and mass spectrometric analyses. The remaining five subunits (FCP-1, FCP-10, FCP-17, FCP-18, and FCP-21) were assigned to *Fragilariopsis cylindrus*-13194[38], *Thalassiosira pseudonana* (T.p.)-Lhcr14[36], *Phaeodactylum tricornutum* (P.t.)-Lhcf4[37], P.t.-Lhcf4, and *C. gracilis* (C.g.)-10219 (Supplementary Table 1), respectively, as the exact sequences matching these structures were not found in the transcriptome sequences of *C. gracilis* and these selected sequences best fit with the cryo-EM map. Sequence analyses showed that the 24 FCPIs belong to four groups, namely Lhcf, Lhcr, Lhca and predicted FCPIs (Supplementary Fig. 5b and Supplementary Table 1)[20,29], and no subunits from the Lhcx group were identified.

Two pair of residues (Glu46-Arg175 and Glu170-Arg51) form salt bridges in all FCPIs to stabilize the structures of FCPI (Supplementary Fig. 5a), which is similar to those of LHCs and FCPIIs[9,10,21,22,41]. The sequences and structures of helices A and B are highly conserved, whereas those of the N-terminal loops, helix C, and other loop regions are not conserved (Fig. 4a and Supplementary Fig. 5a). Based on the structures of the 24 FCPIs and comparisons with Lhcr of red algae and Lhcf from *P. tricornutum*[8,21], they can be divided into five groups. Group I

includes FCPI-4/5/6/7/8/9/10/11/13, most of which were Lhcr family proteins (except FCPI-9, which is a predicted FCP), and their structures are similar with red algal Lhcr[8]. However, FCPI-5 possesses a hydrophilic helix D similar to that found in Lhca from higher plants (Supplementary Figs. 3 and 5a). The structures of FCPI-7/11 can be overlapped almost completely with that of Lhcr1 from red algae, except the short C-terminal loop (Supplementary Fig. 3). FCPI-4/5/6/9/10/13 share similar longer helix C than that in Lhcr1, but is similar to that of Lhcr2 and Lhcr3 proteins (Supplementary Fig. 3). However, helix C of FCPI-8 has an orientation deviated from other Lhcr proteins, in agreement with the phylogenetic analyses. FCPI-9/13 have similar, extra loop regions in the N-terminal (Fig. 4a and Supplementary Fig. 3), whereas longer C-terminal loops are found in FCPI-4/8, which participate in interactions with the PSI core or adjacent FCPI (Fig. 3b–d). Eight FCPI subunits except FCPI-13 in this group are located in the innermost FCPI layer with direct connections to the PSI core (Fig. 1a, b), suggesting that the Lhcr-type proteins are the main peripheral antennas not only binding tightly to the core but also bridging other FCPI subunits with the core.

The second group is an Lhcf group (including some predicted FCPs) consisting of FCPI-2/3/12/15/17/18/19/20/22 and share similar helices and loop structures as that of Lhcf4 from *P. tricornutum* (Supplementary Fig. 3)[21]. FCPI-2 and FCPI-3 are located in the innermost layer and connected with PsaL of the PSI core; FCP-22 is located in the outermost layer, whereas other Lhcf-type subunits are located in the middle layer (Fig. 1a, b). All Lhcf-type antennas are in monomeric state, which differs from the FCPII-A tetramer or FCPII-BC dimer related with the PSII core[21–23]. No phosphorylation of these Lhcf proteins were found, excluding the possibility that some of them may be moved from PSII due to state transition[42]. This suggests that they are intrinsic antennas of diatom PSI.

Group III FCPI contains FCPI-14/16 in the middle layer whose long N-terminal and B–C loops were found to interact with other FCPIs in the outermost layer (Supplementary Fig. 3). FCPI-14

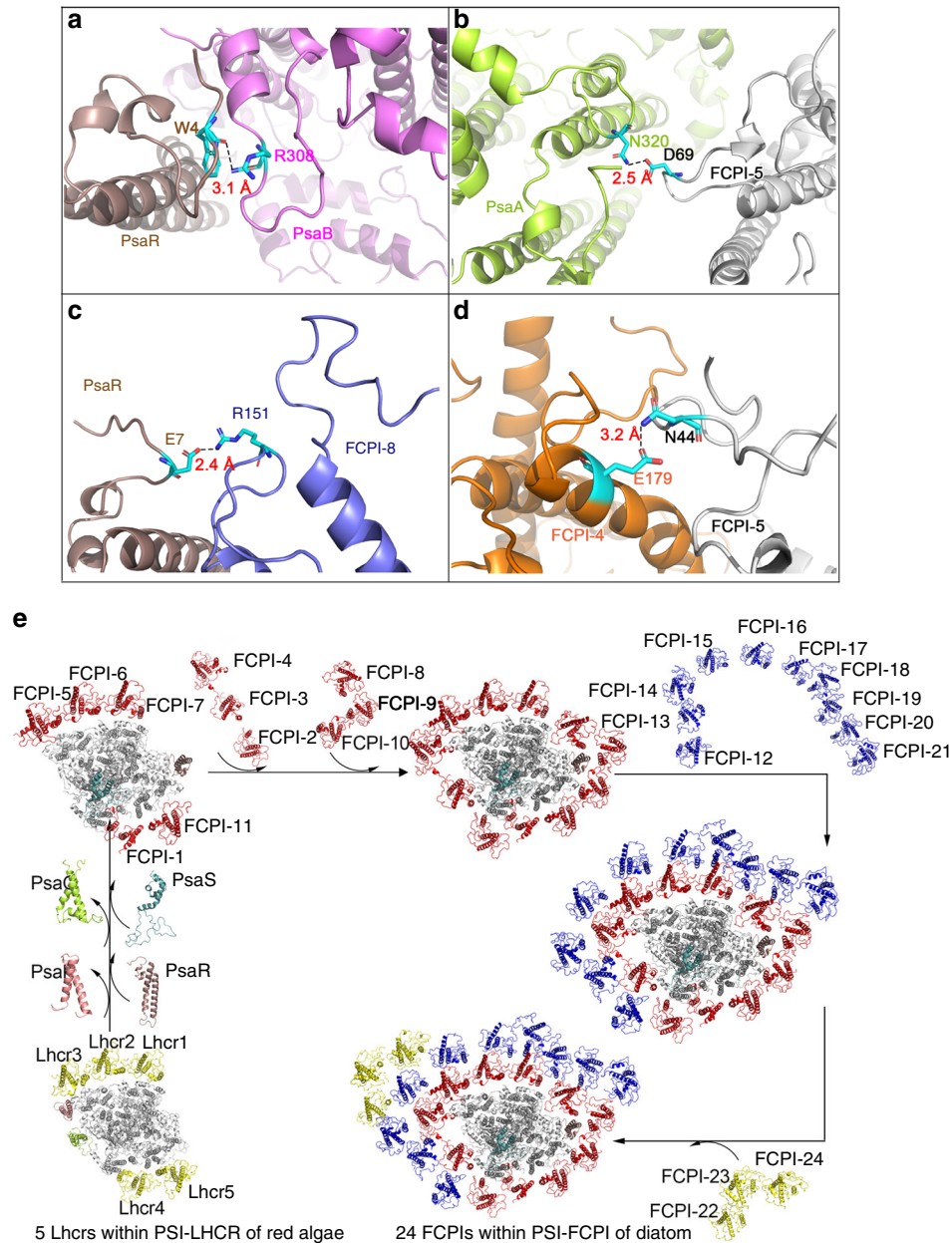

**Fig. 3 Interactions among different subunits and evolutionary development of diatom PSI-FCPI. a** Interactions between PsaR and PsaB. **b** Interactions between PsaA and FCPI-5. **c** Interactions between PsaR and FCPI-8. **d** Interactions between FCPI-4 and FCPI-5. Red number show the distances in Å. **e** Possible development of subunits from the red algal PSI-LHCR to the diatom PSI-FCPI.

uses its N-terminal loop to attach to FCPI-13 at the lumenal surface and its long B-C/C-A loops to fix FCPI-22 in the outermost layer at the stromal surface. Similarly, FCPI-16 uses its N-terminal loop to connect FCPI-5 in the innermost layer and FCPI-24 in the outermost layer. The FCPI-14/16 subunits are connected by FCPI-15 in the middle layer (Fig. 1a, b).

FCPI-21/23/24 are classified into group IV, all of which have distinct hydrophilic helix at the beginning of the N-terminal loop and three long loop structures connecting three transmembrane helices and extended C-terminal regions (Supplementary Fig. 3). Their extended C-terminal loops span across the interfaces of helices A/B at the lumenal surface and approach to helix C, providing interactions with the adjacent FCPIs (Fig. 1 and Supplementary Fig. 3). Although FCPI-21 is in the middle layer and FCPI-23/24 are in the outermost layer, and are therefore all distant from the PSI core, their specific long loop structures

enable them to bind to other FCPIs strongly and thereby stabilizing the whole PSI-FCPI supercomplex.

The remaining FCPI-1 belongs to group V and is located between FCPI-2/11 in the innermost layer (Fig. 1a, b). Its helices A/B are more titled against each other, leading to a more parallel arrangement between Ddx 303 and Fx 305 than those in other FCPs (Fig. 4b and Supplementary Fig. 3). It has an additional B–C loop interacting with PsaB/I/L at the lumenal surface. The seven Chls and four Cars of FCPI-1 are greatly different from those in Lhcr/Lhcf-type FCP antennae, indicating that this is a different type of FCPI-1 antenna and may have previously unidentified functions of light-harvesting and photoprotection.

**Interactions of FCPI with the PSI core**. The high diversities of FCPIs described above implicate a high heterogeneity of the

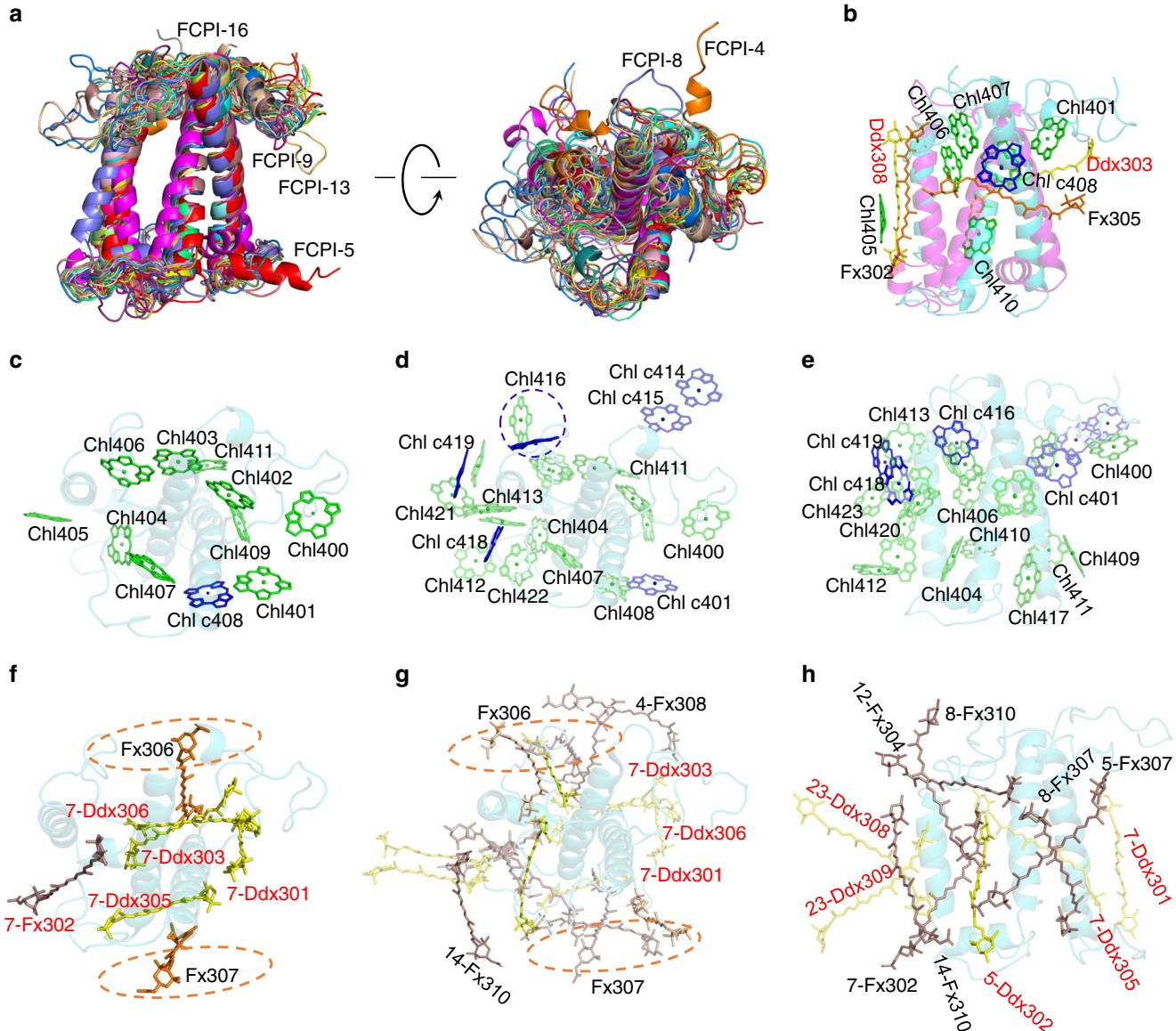

**Fig. 4 Structures and locations of pigments of FCPIs. a** Superposition of the structures of 24 FCPIs. **b** Abnormal FCPI-1 (magenta) structure superposed with the structure of FCPI-7 (cyan). The pigments of FCPI-1 were labeled. Large deviations of protein structures between FCPI-1 and FCPI-7 (Lhcr1 protein) induce large shift of the pigments in FCPI-1. **c** Top view of typical Chl sites in FCPI-7 (Lhcr1 protein) from stromal side. **d, e** Numbering of the 24 Chl sites of diatom FCPIs with the top view (**d**) and side view (**e**), respectively. Green and blue Chls are Chls *a* and Chls *c*, respectively. The blue cycle in **d** indicates the Chl 416 site that can coordinate both Chl *a* and Chl *c* (see Supplementary Table 2). Different colors of Chl 408/401 also indicate these two sites can coordinate both Chl *a* and Chl *c* (Supplementary Table 2). **f** Top view of typical Car sites in FCPI-7 (Lhcr1 protein) from stromal side. The Fx306/307 sites typically seen in Lhcf-type antennae[8,21] are highlighted by circles. **g, h** Numbering of Car sites of diatom FCPIs with top view (**g**) and side view (**h**), respectively.

antenna system that are assembled into a huge and complicated PSI-FCPI supercomplex. As diatoms stem from secondary symbiosis of red algae[43], the five FCPI subunits (FCPI-1/5/6/7/11) in the innermost layer remained in similar positions as Lhcrs in the red algal PSI-LHCI[8] (Fig. 1 and Supplementary Fig. 4a). The binding of other six FCPI subunits in the innermost layer is made possible by changes in the PSI core subunits. FCPI-3 takes the position of the red algal PsaO, which is lost in the diatom PSI, and strongly interacts with the PSI core by hydrophobic interactions and hydrogen bonds. Extensive amino acid residue/pigment interactions are found between FCPI-3 and PsaL, and similar interactions are seen between FCPI-1/10/11 and the PSI core (Supplementary Fig. 6h). The position of PsaK in other PSI core is occupied by FCPI-4, and three to four lipids were found to fill the

cavity between FCPI-4 and PSI core, and connect FCP-3/4 with PsaA (Fig. 1b and Supplementary Fig. 6i). The loss of PsaK also influences the locations of FCPI-4 and FCPI-5: a hydrogen bond between FCPI-4/5 and another hydrogen bond between FCPI-5 and PsaA ensure these Lhcr proteins to bind to the PSI core strongly (Fig. 3d). The previously unidentified PsaR subunit mediates the interactions between PsaB and three FCPI-8/9/10 antennas in the innermost layer (Fig. 3b and Supplementary Fig. 6j). Finally, some lipids were identified between the PSI core and innermost layer FCPIs, as well as among the adjacent FCPIs, which may stabilize the whole structure.

The semi-cycled ring of the middle layer FCPIs covers the PsaA/J/R side, leaving the PsaB/I/L/M side covered with only the innermost FCPI layer (Fig. 1). FCPI-12/13/14/15/16 in this layer

are attached to the FCPI-3/4/5 side and FCPI-17/18/19/20/21 are connected to the FCPI-6/7/8/9/ side. There is a gap between FCPI-16 and FCPI-17 in this layer. The interactions among adjacent FCPIs are similar to those among LHCIs found in PSI-LHCI supercomplexes and consist of hydrophobic and hydrophilic interactions between proteins–proteins, proteins–pigments, and pigments–pigments[9–13] (Fig. 1). An extra Chl $c414–c415$ pair and some Chl/Fx molecules were found to stabilize interactions between FCPI-4 in the innermost layer and FCPI-13 in the middle layer (Supplementary Fig. 6n).

FCPI-22/23/24 are located at the outermost layer in the PsaA side. Additional Fx and Ddx molecules were found in these FCPIs that enhance interactions among adjacent FCPIs (Supplementary Fig. 6o–q and Supplementary Table 2). In total, these 24 FCPIs assemble into an asymmetric belt surrounding the PSI core and no dimeric or tetrameric FCPIs are found.

Based on the structure of PSI-FCPI revealed, the attachment of FCPIs to the PSI core may follow the following steps (Fig. 3e). First, five FCPIs replace the position of five Lhcrs in red algal PSI-Lhcr. Loss of PsaO/K and appearance of the previously unidentified PsaR subunit allow the attachment of additional six FCPIs to the core, forming the innermost layer (Fig. 3e). The innermost layer FCPIs provide the attaching sites for the middle layer FCPIs, which consists of two separate belts FCPI-12/13/14/ and FCPI-17/18/19/20/21, respectively. The FCPIs within these two belts are connected by head to tail, and they attach to the innermost FCPI layer at the FCPI-3 and FCPI-8 side, respectively. Two remaining individual FCPI-16 and FCPI-15 are connected to FCPI-5 and FCPI-4 directly. Finally, FCP-22/23/24 attach to the FCPI-14/15 side at the outermost layer and they bind more Fx and Ddx molecules that may strengthen the interactions with the middle layer FCPIs (Supplementary Fig. 6o–q).

**Pigment binding in the FCPI antennas**. The number of Chls bound to FCPIs varies between 6 and 15, among which 7–12 are Chls $a$ and 1–3 are Chls $c$ (Supplementary Tables 1 and 2). Comparing with the PSI-FCPI structures with 16 antennae[39], fewer Chls $c$ are identified in our FCPI antennae. This is because of the slightly higher resolution of the present structure, where the phytol tail of Chl $a$ was more clearly visible, so they were assigned as Chl $a$, whereas in the previous study[39], the Chls without the phytol tail are all assigned as Chl $c$. The total number of Chl $a$ and $c$ in our FCPIs are 232 and 34, giving rise to a Chl $a/c$ ratio of 6.82, which is much higher than the ratio of 2.37 in the 16 FCPI antennae[39]. The average number of Chl $c$ assigned in our FCPIs structure is 1.4 per FCPI, indicating weaker absorptions of FCPI than the dimeric and tetrameric FCPII at 465 nm[8,21]. The number of Car (Fx and Ddx) bound to FCPIs ranges within 4–9. This gives rise to an average Chl/Car ratio of 1.94, which is slightly higher than 1.67 found in FCPII[22,23], but lower than 2.44 found in the red algal LHCI[8], and much smaller than the average Chl/Car ratio of 4.04 for LHCI/LHCII from higher plants and 4.39–4.53 for green algal LHCI (4.39–4.53)[9–13]. This indicates a fewer number of Fxs in FCPI than that in FCPII, implying a slighter lower capacity of blue-light absorption by FCPI. However, the content of Car in FCPI is much higher than those in LHCI, implying its important role in both energy harvesting and dissipation in the diatom antenna system[44]. In particular, FCPI-23/24 binds 13 Chls and 9 Cars, respectively, giving rise to a much higher Car content (Supplementary Tables 1 and 2). These two FCPIs are located at the outermost FCPI ring, suggesting their possible roles in light harvesting under dim light or facilitating energy dissipation to prevent photodamage.

In most of FCPIs, 9 Chl-binding sites (Chl 401–409) are conserved (Supplementary Table 2); they are located close to 2 cross helices A and B, and are also largely similar to those in Lhcfs, Lhcas, Lhcbs, and Lhcrs (Fig. 4b–e, Supplementary Fig. 6, and Supplementary Table 2), indicating their common and important roles. Twelve Chl-binding sites (Chl 400–411) of diatom Lhcr subunits are overlapped with those of red algal Lhcrs (Supplementary Fig. 6a and Supplementary Table 2); however, additional 12 Chl sites are different among different FCPIs (Supplementary Fig. 6d–f and Supplementary Table 2). Chls 416 and 421 are mixed sites for either Chl $a$ or Chl $c$ in different FCPIs. Chl $a417$ bound to the C-terminal loop region of FCPI-3 and the remaining Chls ($a412$, $a413$, $a416$, and $a421–423$) are close to helices C, B–C, or C–A loop regions (Fig. 4c–e and Supplementary Fig. 6d–f). An extra Chls $c414–c415$ pair was found in the extended N-terminal region of FCPI-4. A previously unidentified Chls $c416$ was found in FCPI-15/16/20/22 and two other previously unidentified Chls $c418$ and $c419$ were found in FCPI-14 and FCPI-16, respectively. Chls $c416/418/419$ were located in the flexible helix C and loop regions (Supplementary Fig. 6f and Supplementary Table 2). All these additional Chls are located at the interfaces between FCPIs at stromal side, suggesting their possible role in mediating energy transfer among adjacent FCPs.

There are also a large variety in the number and binding sites for Cars among FCPIs (Fig. 4g, h and Supplementary Fig. 6k–q). Five Cars (301, 302, 303, 305, and 306 sites) located in the central region of red algal LHCI are conserved in most FCPIs (e.g., see Supplementary Fig. 6k, l for FCPI-7). Among these five Cars, 303 and 305 are in the same location as luteins in LHCs (Supplementary Fig. 6l)[9,10,41]. The remaining Cars are located in the peripheral regions of the apo-protein and show a large variety among different FCPIs, Lhcrs, and LHCs (Fig. 4g, h and Supplementary Fig. 6l).

The central Fxs 303 and 305 are located in a hydrophobic environment, whereas the other peripheral Fxs have one end group extended to stromal surface and another end group inserted into the membrane. The polar environments for these Fxs impose solvent effects to make their absorption red-shifted, which therefore help harvest green light[21]. Fx304 was found only in FCPI-12 and is similar to that in a Lhcf dimer[21]. It is horizontally embedded at the stromal surface, making it suitable to harvest green light (530–570 nm) (Fig. 4h). Moreover, most of the Fx306 are close to Chl 403, which is a Chl $a$ in most FCPIs. The hydrophobic Chl $a$ may make Fx306 a blue Fx (Fig. 4g and Supplementary Table 2). Fx307 is close to Chl $c408$ (Fig. 4d, g), making it a "green" Fx similar as that in the Lhcf dimer[21].

It should be noted that, although the assignment of Chls and Cars in FCPI subunits in the present study is mostly consistent with those of the previous study[39], there are some differences (Supplementary Table 3). Because of the current resolution, it is natural that there may be some different assignments of Fx and Ddx in the two different structures, giving the small differences of the structures of the two types of Cars. Regarding the Chls, only slight differences were observed in some of the FCPI subunits (except the different assignment of Chls $a$ and $c$, see above) (Supplementary Table 3 and Supplementary Fig. 7). FCPI-11 in the current structure has two Chls more than the corresponding FCPI-1 in the structure of Nagao et al.[39], and FCPI-14, FCPI-16 of the current structure have one more Chl, respectively, than the corresponding FCPI-11, FCPI-13 in the structure of Nagao et al.[39]. These Chls are located at the periphery region and thus have no significant effects on the energy transfer pathways (Supplementary Fig. 7). On the other hand, FCPI-6, FCPI-7 in the structure of Nagao et al.[39] have one more Chl, respectively, than the corresponding FCPI-6, FCPI-5 in the current structure. These Chls are located among the bulk Chls and could affect the energy transfer paths; however, they may be mis-assigned in the previous

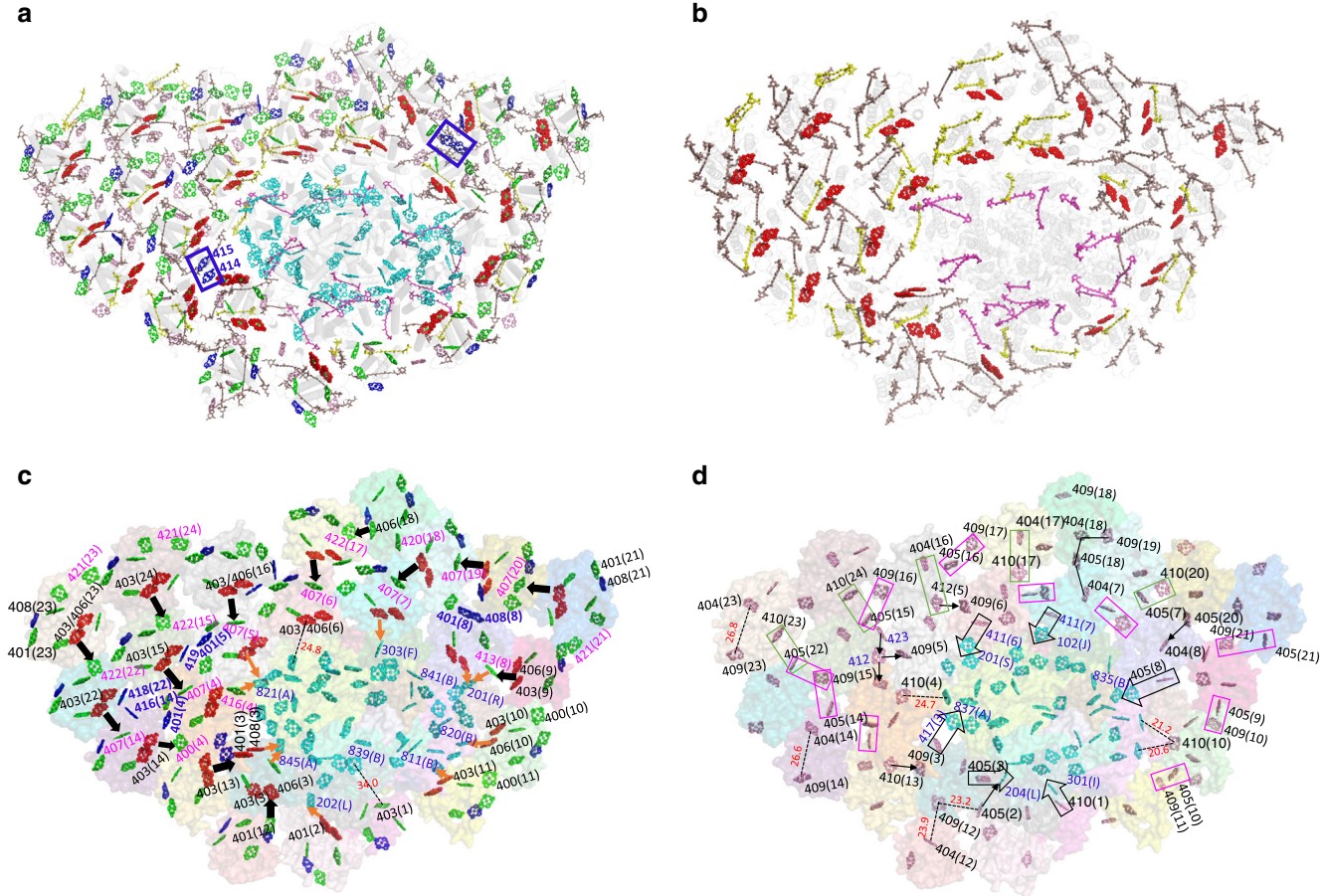

**Fig. 5 Pigment arrangement and energy transfer pathways in the diatom PSI-FCPI. a** Distribution of all pigments in the PSI-FCPI supercomplex. Chls *a* in FCPIs, Chls *a* in the core, Chls *c*, Fx, Ddx, and Bcr are shown in sticks and colored in green, cyan, blue, dark salmon, yellow, and magenta, respectively. Special Chl *a*–*a* pairs are colored in red, two extra Chl *c*–*c* pairs are labeled and placed into blue rectangles. **b** Distribution of all Cars and special Chl *a*–*a* pairs in PSI-FCPI. **c** Energy transfer pathways within PSI-FCPI at the stromal side. Orange arrows indicate EET pathways from inner FCPIs to the PSI core and black arrows indicate EET pathways from outer to inner FCPIs. **d** EET pathways at the lumenal side. Wide arrows indicate EET pathways from inner FCPIs to the PSI core. EET via Chl 405–409 pathways are labeled in magenta squares and those via Chl 410–405 and 410–404 pairs are labeled in orange squares. Other EET pathways associated with Chl 410–409 or other Chls are indicated by thin arrows. In **c** and **d**, labeled Chls depict Chls involved in EET between subunits.

structure based on the low-resolution map[39]. Finally, the number of Chls in FCPI-4 of the current structure was the same as the corresponding FCPI-8 in the previous structure (Supplementary Fig. 7), but it was mis-listed as one Chl less in the previous paper[39].

**Excitation energy transfer in PSI-FCPI.** In total, 515 pigments were found in the PSI-FCPI supercomplex (Fig. 5 and Supplementary Table 1), which form an extremely complicated network ensuring light-harvesting and electron transfer. In FCPIs, all Cars (Fxs and Ddxs) have small distances with Chls (Fig. 5a, b), a feature also found in FCPIIs[21–23], suggesting efficient excitation energy transfer (EET) from Cars to Chls or energy quenching. Twenty-one coupled Chl *a*403–*a*406 pairs are identified in 21 FCPIs, except FCPI-1/12/18, and an additional Chl *a*401–*a*408 pair is found in FCPI-3 (Fig. 5b and Supplementary Table 1). Importantly, most FCPIs face to the PSI core at the Chl *a*403–*a*406 side, whereas the Chl *a*401–*c*408 pair is far away from the core. These results indicate that the Chl a403–a406 pair play an important role in mediating EET from FCPIs to the PSI core.

All the Chl *a*–*a* pairs are distributed at the stromal side, which may serve as red-shifted Chls and provide main EET pathways, whereas individual Chls from each FCPI may provide some possible EET pathways at the lumenal side (Fig. 5c, d and Table 2). The EET pathways are classified into two types: one is the inner pathways from inner FCPIs to the PSI core and the other one is the outer pathways from outer to inner FCPIs (Fig. 5c, d and Table 2). FCPI-5/6/7 share similar EET pathways mediated by PsaJ and PsaF as those of Lhcr1–3 of red algae due to their similar structures and locations (Supplementary Fig. 6g), but lack the pathways mediated by PsaK in red alga and PsaN in higher plants[9]. Most inner pathways are highly efficient due to their short Chl distances at around 11 Å, such as FCPI-8/Chl *a*406-PsaR/Chl *a*201 (center–center distance, 11.0 Å) and FCPI-3/ Chl *a*405-Chl PsaL/*a*204 (10.7 Å) (Fig. 5c, d and Table 2). However, four of these inner pathways (via FCPI-1 and FCPI-6 at stromal side, and via FCPI-4 and FCPI-10 at lumenal side) may be less efficient because of their larger distances (>20 Å). These less-efficient pathways may be compensated for by EET between the stromal and lumenal layers of FCPI-1/6 and FCPI-4/8 (Supplementary Fig. 6h–j). In addition to the Chl *a* pairs, some specific Chls among the inner pathways, such as Chl *a*410, *a*411, and *a*413, *a*416 may also play important roles in mediating the inner EET pathway (Fig. 5c, d and Supplementary Fig. 6h–j).

In the outer EET pathways, a number of coupled Chl *a*–*a* or Chl *a*–*c* pairs were found at pivot positions at the stromal side,

**Table 2 The energy transfer pathways from FCPIs to the PSI core.**

| Code | Pathway | Pigments | Mg–Mg distance (Å) | Location |
|---|---|---|---|---|
| 18-17-6-AL | FCPI-18→FCPI-17 | Chl $a406_{18}$–$a405_{17}$ | 14.7 | Lumenal |
| | FCPI-17→FCPI-6 | Chl $a404_{17}$–$a412_6$ | 18.9 | |
| | FCPI-6→PsaA | Chl $a411_6$–$a817_{PsaA}$ | 14.2 | |
| 18-7-JL | FCPI-18→FCPI-7 | Chl $a405_{18}$–$a404_7$ | 20.4 | Lumenal |
| | FCPI-7→PsaJ | Chl $a411_7$–$a102_{PsaJ}$ | 11.8 | |
| 19-8-BL | FCPI-19→FCPI-8 | Chl $a405_{19}$–$a409_8$ | 18.0 | Lumenal |
| | FCPI-8→PsaB | Chl $a405_8$–$a834_{PsaB}$ | 19.4 | |
| 20-8-BL | FCPI-20→FCPI-8 | Chl $a405_{20}$–$a404_8$ | 18.9 | Lumenal |
| | FCPI-8→PsaB | Chl $a405_8$–$a834_{PsaB}$ | 19.4 | |
| 21-9-10-BL | FCPI-21→FCPI-9 | Chl $a405_{21}$–$a409_9$ | 21.1 | Lumenal |
| | FCPI-9→FCPI-10 | Chl $a409_9$–$a409_{10}$ | 9.5 | |
| | FCPI-10→PsaB | Chl $a410_{10}$–$a410_{PsaB}$ | 14.4 | |
| 11-BL | FCPI-11→PsaB | Chl $a409_{11}$–$a815_{PsaB}$ | 22.2 | Lumenal |
| 1-IL | FCPI-1→PsaI | Chl $a410_1$–$a301_{PsaI}$ | 11.6 | Lumenal |
| 2-IL | FCPI-2→PsaI | Chl $a405_2$–$a204_{PsaI}$ | 21.4 | Lumenal |
| 13-3-IL | FCPI-13→FCPI-3 | Chl $a410_{13}$–$a409_3$ | 16.1 | Lumenal |
| | FCPI-3→PsaI | Chl $a408_3$–$a204_{PsaI}$ | 10.7 | |
| 24-15-4-AL | FCPI-24→FCPI-15 | Chl $a413_{24}$–$a405_{15}$ | 15.5 | Lumenal |
| | FCPI-15→FCPI-4 | Chl $a412_{15}$–$a412_4$ | 13.0 | |
| | FCPI-4→PsaA | Chl $a413_4$–$a818_{PsaA}$ | 24.7 | |
| 21-9-8-RS | FCPI-21→FCPI-9 | Chl $a421_{21}$–$a401_9$ | 9.8 | Stromal |
| | FCPI-9→FCPI-8 | Chl $a403_9$–$a400_8$ | 17.9 | |
| | FCPI-8→PsaR | Chl $a406_8$–$a201_{PsaR}$ | 11 | |
| 11-BS | FCPI-11→PsaB | Chl $a403_{11}$–$a811_{PsaB}$ | 13.0 | Stromal |
| 2-LS | FCPI-2→PsaL | Chl $a406_2$–$a401_{PsaL}$ | 17.6 | Stromal |
| 12-3-AS | FCPI-12→FCPI-3 | Chl $a401_{12}$–$a406_3$ | 16.9 | Stromal |
| | FCPI-3→PsaA | Chl $a407_3$–$a845_{PsaA}$ | 12.5 | |
| 13-4-AS | FCPI-13→FCPI-4 | Chl $a407_{13}$–$a400_4$ | 17.2 | Stromal |
| | FCPI-4→PsaA | Chl $a416_4$–$a821_{PsaA}$ | 13.5 | |
| 14-4-AS | FCPI-14→FCPI-4 | Chl $a406_{14}$–$a400_4$ | 17.7 | Stromal |
| | FCPI-4→PsaA | Chl $a416_4$–$a821_{PsaA}$ | 13.5 | |
| 22-15-5-AS | FCPI-22→FCPI-15 | Chl $a411_{22}$–$a401_{15}$ | 14.6 | Stromal |
| | FCPI-15→FCPI-5 | Chl $a406_{15}$–$a400_5$ | 21 | |
| | FCPI-5→PsaA | Chl $a403_5$–$821_{PsaA}$ | 17.2 | |
| 16-6-AS | FCPI-16→FCPI-6 | Chl $a406_{16}$–$a401_6$ | 18.7 | Stromal |
| | FCPI-6→PsaA | Chl $a403_6$–$a812_{PsaA}$ | 25.7 | |

The excitation energy transfer from FCPs to the core is mainly realized by the chlorophyll and carotenoid molecules between antenna and core. By measuring, comparing and analyzing the pigment arrangement in PSI-FCPI, we found 18 energy transfer pathways in diatom PSI-FCPI: 18-17-6-AL, 18-7-JL, 19-8-BL, 20-8-BL, 21-9-10-BL, 11-BL, 1-IL, 2-IL, 13-3-IL, 24-15-4-AL, 21-9-8-RS, 11-BS, 2-LS, 12-3-AS, 13-4-AS, 14-4-AS, 22-15-5-AS, and 16-6-AS. The numbers represent FCPs, the middle uppercase letter represents core subunit, and the last uppercase letter S represents Stromal side and L represents lumenal side. The pathways were based on the distances of Mg–Mg in Chl $a$ around adjacent FCPs.

which is in analogy to LHCs and FCPIIs[9,10,21–23,41]. FCPI-18/23/24 in the outermost layer transfer energy via their Chl 403–406 pairs to Chl $a$422 of FCPI-15/17/22 in the middle layer at the stromal side (Fig. 5c), and FCPI-4/5/6/7-Chl $a$407, FCPI-4-Chl $a$400, FCPI-8-Chl $a$413 in the innermost layer may accept energy from the middle layer (Fig. 5c and Table 2). In addition, two Chl $a$–$a$ pairs in FCPI-3 may accept energy from FCPI-12/13 in the middle layer and transfer them to the PSI core (Fig. 5c). FCPI-23/24 may share their energy via Chl $a$421. Significantly, all Chls $c$ appear to harvest and transfer light energy at the stromal side. Except for coupled Chls $c$ with Chls $a$, two Chl $c$–$c$ pairs (Chl $c$401–$c$408 and Chl $c$414–$c$415) are located in the innermost layer and can also mediate EET efficiently. The individual Chls $c$ (416, 418, and 419) are located close to special Chl $a$–$a$ pairs, suggesting efficient EET between Chls $a$ and $c$[21].

At the lumenal side, outer EET pathways are possible among Chl 405–409, 405–410, 404–410, 410–405, and 410–409 (Fig. 5d and Table 2). FCPI-19/Chl 409-FCPI-18/Chl 404, 405-FCPI-17/Chl 404 form a possible EET pathway from outer to inner FCPIs. Most outer FCPIs have three o four Chls at the lumenal side with large inter-distances and they may share their energy with the stromal side Chls via the middle Chls $a$409–$a$402. FCPI-4/Chl $a$412 and FCPI-15/Chl $a$423 have a shorter distance, and

therefore may accelerate EET between FCPI-15 and FCPI-4 (Fig. 5d and Table 2).

In conclusion, the diatom PSI-FCPI supercomplex possesses a huge antenna system to expand its cross-section for light harvesting by incorporating 24 FCPI subunits with a great variety in terms of both protein sequences and pigment binding. As a result, FCPI represents the largest antenna system among all of the PSs reported so far. This is apparently a result of adaptation to the fluctuating light conditions under water that diatoms experience. Therefore, the huge antenna system, together with the unique pigments and their binding environments, provide a highly efficient energy harvesting and dissipation system to assure the success of diatoms in the aqueous environment.

## Methods

**Cell culture and PSI-FCPI purification**. Cells of *C. gracilis* (Center for Collections of Marine Bacteria and Phytoplankton, Xiamen University, Xiamen, China) were grown in F/2 medium supplemented with artificial sea water and bubbled with air containing 2% $CO_2$, at 22 °C under continuous light (40 µmol photons $m^{-2} s^{-1}$) for 6–8 days. The cells in the late logarithmic phase were collected by centrifugation ($7000 \times g$, 10 min) and resuspended in an ice-cooled buffer MMC [50 mM 2-morpholinoethanesulfonic acid, monohydrate-NaOH pH 6.0, 10 mM $MgCl_2$, 5 mM $CaCl_2$][33,34]. The collected cells were partially broken by freezing and were stored at −80 °C until use.

The diatom cells were thawed at room temperature and then disrupted by a high-pressure chamber, and the unbroken cells were removed by centrifugation at $5000 \times g$ for 10 min. Thylakoid membranes were collected by centrifugation at $4000 \times g$ for 10 min at 4 °C. The thylakoid membranes were resuspended in the buffer MMC at 1 mg Chl $a$ per mL and solubilized with 1.0% (w/v) $n$-dodecyl-β-D-maltopyranoside (DDM) (Anatrace, Maumee, OH) for 10 min on ice. The thylakoid membranes were centrifuged at $40,000 \times g$ for 15 min to remove unsolubilized materials and the supernatant was loaded onto a linear sucrose density gradient $(0.3 - 1.1$ M sucrose) in the buffer MMC containing 0.02% β-DDM followed by centrifugation at $250,000 \times g$ for 18 h. After centrifugation, two major bands and two minor bands were obtained (Supplementary Fig. 1a). The upper major, brown band is FCPs and the lower major band is PSI-FCPI. The two minor green bands between the two major bands contain PSI cores and PSII cores, respectively. The second major band was collected and its sucrose was removed by gel filtration chromatography in the buffer MMC containing 0.02% β-DDM. The peak eluted was collected and concentrated using a membrane filter (molecular weight cut-off: 100 kDa AMICON). All procedures were performed under dim green light at 4 °C or on ice.

**Pigment analysis.** Pigments from PSI-FCPI were extracted with cold 90% acetone (V/V) and kept at 4 °C overnight in the dark. After centrifugation, the supernatant was collected and used to determine the pigment concentration by high-performance liquid chromatography (HPLC)[21,45]. The HPLC column used was a C-18 reversed-phase column (5 μm, 100 Å, 250 × 4.6 mm, Grace, USA) equipped in a Waters e2695 separation module with a Waters 2998 photodiode array detector. The pigments were eluted at 20 °C at a flow rate of 1 ml/min with the following steps: 0–20 min, linear gradient of buffer A (methanol : water = 90 : 10) from 100% to 0; 20–22 min, 100% buffer B (ethyl acetate); 22–23 min, linear gradient of buffer B from 100% to 0; and 23–28 min, 100% buffer A. The elutes were detected at 445 nm with a wavelength sweep surface range of 300–800 nm.

**Peptide and sequence analyses of the PSI-FCPI from C. gracilis.** Protein composition was analyzed by SDS-polyacrylamide gel electrophoresis with 16% polyacrylamide and 7.5 M urea[46]. Each band was cut from the gel and used for matrix-assisted laser desorption ionization time-of-flight (MALDI-TOF) mass spectrometry analysis. For analyzing the transcriptome sequences of C. gracilis, the total RNA was extracted with a TRIzol Reagent according to the manufacturer's instructions (Invitrogen, CA, USA)[22]. Sequencing libraries were generated using NEBNext® Ultra™ RNA Library Prep Kit for Illumina® (NEB, MA, USA) following the manufacturer's recommendations. The first-strand cDNA was synthesized using random hexamer primers and RNase H, and the second-strand cDNA was synthesized using DNA polymerase I and RNase H. Further purification with QiaQuick PCR kits (QIAGEN, Germany) and modifications including terminal repair, A-tailing, and adapter addition were performed. The targeted products were retrieved by agarose gel electrophoresis and PCR was performed to obtain the final cDNA library. Sequencing was performed using Illumina HiSeqTM 2000 according to the manufacturer's instructions (Illumina, CA, USA). Reference-free overlapping and connections were examined to combine the original reads into contigs, which were further assembled into unigenes by means of paired-end assembly and gap filling[22]. Subunits of FCPI were identified by retrieving the MALDI-TOF results from the transcriptome sequences or databases in the National Center for Bio-technology Information. Sequence alignment and phylogentic analysis were performed with MEGA X[47] and Phylogeny.fr[48].

**Spectroscopic measurements.** Room-temperature absorption spectra were measured with a UV-Vis spectrophotometer (Shimadzu, Japan) in the MMC buffer containing 0.02% β-DDM. Fluorescence emission and excitation spectra at 77 K were measured using a fluorescence spectrophotometer (F-7000, Hitachi, Japan) at a Chl concentration of 20 μg Chl per ml in the MMC buffer supplemented with 50% glycerol and 0.02% β-DDM.

**Cryo-EM data acquisition.** An aliquot of 4 μl of concentrated diatoms PSI-FCPI supercomplex was loaded onto a glow-discharged holey carbon grid (Quantifoil Cu R1.2/1.3, 400 mesh). The grid was blotted for 4.0 s and plunged into liquid ethane cooled by liquid nitrogen using the Vitrobot Mark IV (Thermo Fisher) working at 100% humidity and 8 °C. Grids were loaded into the Titan Krios electron microscope (Thermo Fisher) operated at 300 kV and equipped with a spherical aberration (Cs) image corrector and a Gatan Gif Quantum energy filter (slit width 20 eV). In total 8535 movie stacks were collected using a K2 Summit counting camera (Gatan Company) in super-resolution mode with a nominal magnification of ×105,000, yielding a final pixel size of 1.091 Å. Each stack of 32 frames was exposed for 5.6 s, leading to a total dose for a stack of about 50 $e^-$ per Å². The defocus values were set between −1.5 and −2.5 μm. AutoEMation[49] was used for auto-mated data collection. All movie stacks were corrected by MotionCorr2[50] and dose weighting was performed.

**Data processing.** The defocus values and the parameters of astigmatism were estimated by CTFFIND4[51] using the corrected micrographs without dose weighting. Among the collected images, 1839 micrographs whose maximum

resolution was estimated below 5 Å by CTFFIND4 were discarded. A total of 891,804 particles were auto-picked by RELION-3.0[52] from 6696 good micrographs, for which the templates were low-pass-filtered to 20 Å. Two rounds of two-dimensional classification using RELION-3.0 were performed and 695,499 particles were selected for three-dimensional (3D) classification without imposing symmetry. The map of green algae PSI-LHCI[11] (EMDB EMD-9670) low-pass-filtered to 60 Å was used as an initial reference for the 3D classification. A final set of 164,480 particles from 3D classification were kept for high-resolution refinement. To improve the resolution of auto-refinement, particles were re-centered and re-extracted from the dose-weighted micrographs, and per-particles defocus, per-particles astigmatism, and beam-tilt estimation were performed using RELION-3.0. A soft-edged mask was applied in auto-refinement and post-processing, leading to a 2.38 Å density map of the diatoms PSI-FCPI complex. The density map was sharpened by applying a negative B-factor using automated procedures. The resolution was estimated based on the gold-standard FSC 0.143 criterion[35]. The local resolution map was generated by ResMap[53].

**Model building and refinement.** For model building of the diatom PSI-FCPI supercomplex, the structure of red algal PSI-LHCR (PDB: 5ZGB)[8] was first manually placed and rigid-body fitted into the 2.38 Å resolution cryo-EM map with UCSF Chimera[54]. The amino acid sequences were subsequently mutated to counterparts in C. gracilis PSI obtained from transcriptome sequencing and mass spectrometric analyses. Every residue and cofactor were manually checked and adjusted with COOT[55]. De novo model building was performed on the previously unidentified subunit R (chain h in PDB file). The previously unidentified subunit S (chain g in PDB file) was assigned as polyalanines due to the absence of suitable sequences. All FCP monomers were identified and were mutated from a FCPII monomer (PDB: 6A2W)[21], except for FCP-1, FCP-10, FCP-17, FCP-18, and FCP-21, whose sequences could not be found from the transcriptome sequences and were modeled using sequences of F. cylindrus-13194[38], T.p.-lhcr14[36] P.t.-lhcf4[37], P.t.-lhcf4, and C.g.-10219, respectively, as these sequences best fit with the cryo-EM map. The overall model of diatom PSI-FCPI was completed with alternating rounds of manually adjustment in COOT[55]. Finally, real-space refinement was performed using Phenix[56] with geometry and secondary structure restraints. The statistics for data collection and structure refinement are summarized in Table 1 and Supplementary Fig. 2e.

**Reporting summary.** Further information on research design is available in the Nature Research Reporting Summary linked to this article.

## Data availability
Atomic coordinates and cryo-EM map have been deposited in the Protein Data Bank under the accession numbers of EMD-30012 and 6LY5, respectively. The raw electron microscopy images used to build the 3D structure are available from the corresponding authors upon request.

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

## Acknowledgements

We thank J. Lei and the staff at the Tsinghua University Branch of China National Center for Protein Sciences (Beijing) for providing facility support on the Cryo-EM and High-Performance Computation platforms. This work was supported by the National Key R&D Program of China (2017YFA0503700, 2017YFA0504600, 2019YFA0906300, and 2016YFA0501101); the National Natural Science Foundation of China (31970260, 31600191, 31861143048, and 31670745); a Strategic Priority Research Program of CAS (XDB17000000), a CAS Key Research Program for Frontier Science (QYZDY-SSW-SMC003), and Youth Innovation Promotion Association of CAS (2020081).

## Author contributions

J.-R.S., W.W., and S.-F.S. conceived the project. C.X., G. Han, X.C., S.Z., Y.Y., and W.W. performed the sample preparation, characterization, and sequence analysis. X.P., Y.H., and G. Huang collected and processed the cryo-EM data. X.P., Y.H., G. Huang, X.Q., and W.W. built and refined the structural model. W.W., C.X., X.P., T.K., S.-F.S., and J.-R.S. wrote the manuscript. All authors discussed and commented on the results and the manuscript.

## Competing interests

The authors declare no competing interests.
