## [Peer Review File · Nature Communications]

REVIEWER COMMENTS

Reviewer #1 (Remarks to the Author):

The manuscript by Xu et al presents the structure of a photosystem I-antenna supercomplex from diatoms, which is the largest complex found so far, including 24 antenna monomers and the accompanying huge amounts of pigments. It is a tremendous work, including the layout of the manuscript that describes the complicated built of the supercomplex and the network of cofactors like chlorophylls and carotenoids in sufficient clarity. It also includes a distance based model for excitation energy transfer into the core of photosystem I. By the details provided, this structure will certainly be the basis for many further publications about light harvesting and photoprotective abilities of diatoms. Due to the possible comparison between different phylogenetic taxa it is also interesting for a broader audience.

The manuscript is accompanied by another manuscript also from Prof Shen and colleagues, reporting on the structure of a photosystem I-antenna supercomplex complex from the same species, which, however, lacks 8 of the 24 antenna subunits. Thus, there are many redundancies between the two texts and amazingly enough also different attributions of densities to cofactors and attribution of proteins to protein families. For the interested reader it would thus be more fruitful if those discrepancies would be outlined and an explanation given why different interpretations were chosen.

Besides this issue, there are some major and some minor concerns as listed below:

Major points

i) Fcp1-4-11 and 13 are shown to be almost identical (group I FCP1s, Extended Figure 3 shows a perfect overlap of the transmembrane helices). However, concerning the arrangement of helices A and B in Fig 1 they fall into 2 groups: FCP1-4, -5, -7, and -13 look identical, but totally different from FCP1-8 – 11. Even worse for group 2, almost every FCP has a different appearance (tilt of helices). If the orientation with respect to the membrane plane would be the same, they would only differ by rotation, but differences are much stronger. So is there a different orientation with respect to membrane plane? Authors should explain this in the text.

ii) PsaS was found as new subunit on the stromal side: authors should comment on possible functions; why would diatoms have the need for a further subunit in that location?

iii) Phylogenetic considerations: "Assembly" of a protein complex is used for putting together subunits after protein synthesis, not for changes during evolution. Here, e.g. "evolutionary development" would be a better term (Fig. 3 and text line 253 onwards).

The tree shown in the extended data set is not up to standard. In Figure 5c (tree) FCP1-1 is named as if it would be a *C. g.* sequence. This is not allowed since the sequence is from *Fragilariopsis*, same for FCP-10 and 17/18 which were taken from *T. pseudonana* and *P. tricornutum*, respectively. One has to use the original sequence and species names and might mention in the legend that probably sequences in *C. g.* are identical or highly similar. Anyway, the selection of sequences for tree building is not up to standard: the lhcf group was taken from *P. tricornutum* (pennate diatom), lhcx group from *T. pseudonana* (centric diatom), lhcr again only from *P. tricornutum* and lhca from both. There are plenty of phylogenetic analyses published, all using as many sequences as there are available, since otherwise trees are heavily biased. So please use all sequence data available for this analysis and describe the attribution in the main text based on a better tree. And what are predicted FCPs?

iv) The manuscript needs some language editing, grammar is often incorrect and there are a lot of typos present (e.g. phylloquinine).

Minor points:

Bcr (β -carotene) seems to be meddled with Car (carotenoid) on several occasions, please check carefully: e.g. line 277, FCP does contain 9 carotenoids, but certainly not 6 β -carotenes as written, line 291, ...

Line 292: this suggests that all these FCPs have three Chl c in addition, please rephrase to make it clear that maximally 2 per protein are present

Line 305: ...are in the same location as luteins in e.g. LHCII...

Line 327: the Chl/Chl pairs mentioned are not labelled together in any of the figures concerning single FCPs (especially 403/406 is missing), this makes it hard to understand. Appropriate would be a more thorough labelling in Figure 3

Figures and legends:

Figure 4 legend seems to be incomplete:

b) pigments of FCP1-7: not shown?

c) cannot present all 24 FCPs since deviations shown in a) are much stronger for the protein backbone. In addition, what do the circles highlight?

d) and e) what are green and blue Chls, respectively? What do the circles highlight? Why are Chl 408/401 coloured differently in d) versus e/f?

g) What do the circles highlight?

Figure 5 c and d: I guess it should read: "Labels depict Chls involved in EET between subunits", since all other Chl should work in EET as well

Extended Material

Figure 1: b) please provide wavelength at which pigments were detected in HPLC, c) on figure : thylakoid (not thylakoid)

Figure 3b: Lhcr1 instead of Lhca1?

Table 2: please format in a way that the first row is legible

Reviewer #2 (Remarks to the Author):

Photosynthesis is a unique energy conversion process performed by plants, algae, and cyanobacteria where photons from the sunlight are converted into chemically fixed energy with a generation of oxygen as a side-product of water splitting. Diatoms being microalgae are widely spread all over the Earth, including water and soil habitats, produce annually up to half of all planet oxygen and ocean organic compounds.

Thus, the organism in the reviewed research is of interest to a wide circle of scientific society. The article revealed the structure of the photosystem I of diatoms.

The photosystem I (PSI) is a large multi-subunit membrane-embedded complex playing the key role in the photosynthetic process. It carries out the light conversion steps that ultimately lead to the production of ATP by ATP-synthase and reduction of NADP to NADPH. In the reviewed article the PSI-FCPI supercomplex of diatoms was resolved at an extremely-high resolution about 2.38 Å, revealing the structural details of a PSI and surrounding FCPI units (PSI-FCPI supercomplex).

It appeared that PSI-FCPI supercomplex is very big and possesses similarities in structural organizations from both red algae and cyanobacteria. At the time, some unique structural differences, common for diatom, namely for *Chaetoceros gracilis*, in both PSI and FCPI organization, determine its physiology, ecology, and evolution. These differences include the presence or absence of some PSI periphery subunits, which are mainly found at higher plants, cyanobacteria, and red algae, the unique organization of FCPI units surrounding PSI, etc.

The special attention in this article is paid to the FCPI subunits. They are very numbered (24 in total) what makes the PSI-FCPI supercomplex structure exceptional in terms of size and complicity.

Recently the PSII-FCPII supercomplex structure of *C. gracilis* was revealed at 3.3 Å resolution (Nagao et al., 2019). The solved structure shows a huge pigment network, and extensive protein-protein, pigment-protein, and pigment-pigment interactions within the PSII-FCPII supercomplex. In the case of PSI-FCPI supercomplex, the FCPI subunits cannot form higher-order structures (e.g., homotetramer) as it is described in PSII-FCPII (Nagao et al., 2019). At the same time, FCPI subunits in PSI-FCPI supercomplex have original layer organization, surrounding PSI.

There are apparent differences in FCP organization in PSI-FCPI, and PSII-PCPII supercomplexes reflect their different functions during electron transport.

The FCPI subunits in PSI-FCPI supercomplex are monomeric and have original layer organization, around PSI supercomplex.

Thus, this article will bring an inside into the understanding functioning of diatoms electron transport chain.

All in all, I consider the reviewed article is of importance to the scientific community in particular structural biologist and plant physiologist and would assess the work as recommended for the publication in the Nature Communications journal after the appropriate corrections.

At the same time, I would like to point out that:

1. Chl a/c and Fx can absorb light not only in the blue-green part of the visible spectrum (as stated here 67-68) but also in yellow-green one for Fx.

2. The lack of structure/function information of the novel PsaS subunit (130). As we see from the main Fig. 1c and Extended data, Fig. 2b, the PsaS subunit is one of the lowest resolved. The reason can be the flexibility of this subunit, the radiation damage, issues connected to image processing steps.

PsaS subunit is being in the center of PSI-FCPI supercomplex, located near to the core PsaB subunit in the text of the article named as having the 'peripheral location' (132). What does that mean?

3. The representation of the Fig. 1 | overall structure of the PSI-FCPI supercomplex (497) – file name 251291_0_art_file_3393120_q7fkfr.pdf.

In that figure, we see the structure of PSI-FCPI from 1a – stromal, 1b - luminal side. However, on the 1a only the peripheral FCPI are named, whereas the subunits of PSI are only colored and have no name on them. All the PSI-FCPI supercomplex represented as sticks and bounds.

On the 1b the representation is different - FCPIs are numbered, colored in 3 different colors, which probably reflect inner-most / second layer / outer layers of FCPI (no explanation on Fig 1) and shown as a ribbon. The PSI subunits are shown as one grey surface and named. Since we cannot see all the subunits equally (e.g. PsaS) from both stromal and luminal sides, the stromal and luminal sides might be represented consistently, in the same style (e.g., in 1b have a PSI surface separated on subunits with the same color as 1a, etc.). If it is not possible please (due to the space for example) have 2 representations for each side.

4. The separation FCPI on the 3 layers (147) (based on their proximity to the PSI? Not explained) and V groups based on the structures and locations (171) is somewhat confusing.

First, the layers are named as:

- inner-most (148): FCPI-1-11;

- the second layer (149) (sometimes named later as middle, why not when the first time introduced?): FCPI-11(again? I guess should be FCPI-12)-22;

- outer-most (150) / but sometimes out-most (278): FCPI-22-24;

Second, when there is a description of group III FCPI-14/16 (196) the FCPI-14 connects with FCPI-22 from the outer layer and FCPI-13 that named as being from the inner layer (198). I guess there is a misleading, also in Fig. 1b (497-498).

5. PSI-FCPI purification procedure: for the thylakoid solubilization the 0.02% β -DDM was used. Is there substantial evidence that the resulting PSI-FCPI supercomplex contains all the FCPI subunits? You could also try the final solubilization with different β -DDM detergent concentrations in order to see whether PSI-FCPI supercomplex that you get is the only one functional and as close to the native conditions as possible.

6. Did you perform any negatively stained screening (recommended for fast sample evaluation)?

Was PSI-FCPI supercomplex the only one present in the cryo-sample under the used conditions?

7. The PSI-FCPI purification procedure worked at 4°C or on ice (584). What is the certain reason to freeze the sample at 8 C°?

8. 251291_0_extended_data_3393121_q7fkfr.pdf

Extended data Fig. 2b. The local resolution PSI-FCPI supercomplex 3D map looks quite solid. As expected, most of the FCPIs peripheral subunits are lower resolved.

Nonetheless, how can be explained the fact that some of the FCPIs peripheral subunits, all arising from the inner-most layer, namely FCPI 9;10;11 have higher resolution, comparable with the central part of the PSI-FCPI supercomplex? Can it be that those FCPI subunits are not edge-localized under the natural conditions? Can the reason be in the wrong angular assignment of single particles, sample heterogeneity, flexibility, or radiation damage?

You can answer some of those questions, processing the 3D map in MonoRes and MonoDir.

9. Extended data Fig. 2b. –the local resolution 3D map is shown for the luminal side only. Please add the stromal side illustration. Also, add the description accordingly.

10. Extended Data Fig. 2c – Which FSC do we observe? Please add the FSC plot the will have FSC corrected, unmasked, masked (you can get this information from relion 3.0 that you used for the postprocessing).

11. Extended Data Fig. 2c – Why do FSC plot has a small gap in the low range of frequencies? What determines that?

12. Why wasn't the Bayesian polishing step performed after the particle CTF parameters refinement and correction of estimated beam tilt?

13. Please add angular distribution plot for the final reconstruction, so that we could estimate angular coverage of the projection sphere.

Responses to the comments of reviewer #1

The manuscript by Xu et al presents the structure of a photosystem I-antenna supercomplex from diatoms, which is the largest complex found so far, including 24 antenna monomers and the accompanying huge amounts of pigments. It is a tremendous work, including the layout of the manuscript that describes the complicated built of the supercomplex and the network of cofactors like chlorophylls and carotenoids in sufficient clarity. It also includes a distance based model for excitation energy transfer into the core of photosystem I. By the details provided, this structure will certainly be the basis for many further publications about light harvesting and photoprotective abilities of diatoms. Due to the possible comparison between different phylogenetic taxa it is also interesting for a broader audience.

The manuscript is accompanied by another manuscript also from Prof Shen and colleagues, reporting on the structure of a photosystem I-antenna supercomplex complex from the same species, which, however, lacks 8 of the 24 antenna subunits. Thus, there are many redundancies between the two texts and amazingly enough also different attributions of densities to cofactors and attribution of proteins to protein families. For the interested reader it would thus be more fruitful if those discrepancies would be outlined and an explanation given why different interpretations were chosen.

Author's answers:

First of all, thank you very much for your positive comments and important suggestions to improve our manuscript. We have considered your suggestions carefully and made revisions accordingly, which are listed below.

The publication (“Structural basis for assembly and function of a diatom photosystem I-light-harvesting supercomplex” Nagao et al. 2020) presents a diatom PSI-FCPI supercomplex with 16 antennae from the same centric diatom (*Chaetoceros gracilis*). Our current results show that the PSI-FCPI supercomplex contains 24 antenna subunits and also achieved higher resolution. One highlight in our manuscript is the higher local resolution of PSI core and the inner FCPI antennae at about 2.2 Å, allowing us to identify some Fx and Ddx in PSI core and more tails of Chl *a*. For example, only 3 Chls *c* were assigned in FCPI-4, which is much more less than the 7 Chls *c* assigned in the corresponding FCPa8 in Nagao et al. 2020. More Chls *a* facilitate EET from FCPIs to PSI core. Another important point is the additional 8 FCPI antennae including some Lhcf-type antennae encircling PSI core, found in the current structure but not in the structure of Nagao et al. 2020. We speculate that the loss of the FCPI subunits was caused by different culture conditions (temperature, light, carbon dioxide concentration, etc.) of diatom cells, and/or during the extraction and purification of the protein complex. We have added some explanations to the revised manuscript to explain this (page 5, lines 149-152).

Besides this issue, there are some major and some minor concerns as listed below:

Major points

i) Fcp1-4-11 and 13 are shown to be almost identical (group I FCP1s, Extended Figure 3 shows a perfect overlap of the transmembrane helices). However, concerning the arrangement of helices A and B in Fig 1 they fall into 2 groups: FCP1-4, -5, -7, and -13 look identical, but totally different from FCP1-8 – 11. Even worse for group 2, almost every FCP has a different appearance (tilt of helices). If the orientation with respect to the membrane plane would be the same, they would only differ by rotation, but differences are much stronger. So is there a different orientation with respect to membrane plane? Authors should explain this in the text.

Author's answers:

Yes, there was a different orientation with respect to the membrane plane in the old Extended Fgi. 3a, because we wanted to show more details. We modify them into the same orientation now and added more explanations in the legend to Supplementary Fig. 3.

ii) PsaS was found as new subunit on the stromal side: authors should comment on possible functions; why would diatoms have the need for a further subunit in that location?

Author's answers:

Based on the structure and location of PsaS, we speculate that it may have some protective effect for Psa C/D/E. We added this in the revised manuscript (page 5, lines 136-138).

iii) Phylogenetic considerations: "Assembly" of a protein complex is used for putting together subunits after protein synthesis, not for changes during evolution. Here, e.g. "evolutionary development" would be a better term

Author's answers:

We modified this word in the legend to Fig. 3 based on your suggestion.

The tree shown in the extended data set is not up to standard. In Figure 5c (tree) FCP1-1 is named as if it would be a *C. g.* sequence. This is not allowed since the sequence is from *Fragilariopsis*, same for FCP-10 and 17/18 which were taken from *T. pseudonana* and *P. tricornutum*, respectively. One has to use the original sequence and species names and might mention in the legend that probably sequences in *C. g.* are identical or highly similar.

Author's answers:

Based on your comments, we modified Supplementary Figure 5a and 5b (tree) to indicate the FCP1 sequences and species that they were taken from. The reason we used species other than *C. g.* for FCP1-1, 10, 17/18 is that, we could not find the corresponding sequences in our transcriptome sequences, so we take the

corresponding sequences from other known species that seem to best fit with the density map we obtained.

-Anyway, the selection of sequences for tree building is not up to standard: the lhcf group was taken from *P. tricornutum* (pennate diatom), lhcx group from *T. pseudonana* (centric diatom), Lhcr again only from *P. tricornutum* and lhca from both. There are plenty of phylogenetic analyses published, all using as many sequences as there are available, since otherwise trees are heavily biased. So please use all sequence data available for this analysis and describe the attribution in the main text based on a better tree.

Author's answers:

We have used all diatom lhc family members queried from NCBI and Uniprot databases in the tree (excluding repetitive sequences) to build the tree. The current results contain most relevant sequences available, and we have modified the names of sequences from *C. gracilis*.

And what are predicted FCPs?

Author's answers:

We identified some PSI-FCPI components by mass spectrometry and found the corresponding sequences in the *C. gracilis* transcriptome. By predicting their three-dimensional structure, we found that they were some FCPs, but the corresponding genomic sequences were not published and named, so we classified them as predictive proteins.

iv) The manuscript needs some language editing, grammar is often incorrect and there are a lot of typos present (e.g. phylloquinine).

Author's answers:

We re-checked the spelling of the text and made changes where necessary, such as changing phylloquinine to phylloquinone (line 100).

Minor points:

Bcr (β -carotene) seems to be meddled with Car (carotenoid) on several occasions, please check carefully: e.g. line 277, FCP does contain 9 carotenoids, but certainly not 6 β -carotenes as written,

Author's answer:

We checked and corrected Bcr and Car in the text to avoid confusions, e.g. lines 292, line 313 and Table 2 of the revised manuscript. Thank you.

line 291, ...Line 292: this suggests that all these FCPs have three Chl *c* in addition, please rephrase to make it clear that maximally 2 per protein are present.

Author's answer:

According to your suggestion, we modified line 291-293 as follows (lines 306-310, revised manuscript):

“An extra Chls c414-c415 pair was found in the extended N-terminal region of FCPI-4. A new Chls c416 was found in FCPI-15/16/20/22, and two other new Chls c418 and c419 were found in FCPI-14 and FCPI-16, respectively. Chls c416/418/419 were located in the flexible helix C and loop regions (Extended Data Fig. 6f, Extended Data Table 3).”

line 305: ...are in the same location as luteins in e.g. LHCII...

Author's answer:

According to your suggestion, we modified line 316-316 as follows.

“Among these five Cars, 303 and 305 are in the same location as luteins in LHCs (Extended Data Fig. 6l)^{9,10,41}”

Line 327: the Chl/Chl pairs mentioned are not labelled together in any of the figures concerning single FCPs (especially 403/406 is missing), this makes it hard to understand. Appropriate would be a more thorough labelling in Figure 3

Author's answer:

We added labels of Chls 403/406 in Fig. 5c and Fig. 4c. Fig. 3 mentioned by the reviewer does not represent the positions of Chls.

Figures and legends:

a) Figure 4 legend seems to be incomplete:

Author's answer:

We added some additional explanations to the legend of Figure 4 to complete it.

b) pigments of FCP1-7: not shown?

Author's answer:

We present Fig.4b to show large deviations of protein structures between FCPI-1 and FCPI-7 (Lhcr1 protein), thus the 7 Chls and 4 Cars of FCPI-1 are greatly different from those in Lhcr/Lhcf-type FCP antennae which indicates a new FCPI-1 type that may have novel functions on light-harvesting and photoprotection.

As the picture shown below, pigments of FCPI-7 and FCPI-1 make big confusions (Chls of FCPI-7 are depicted in bluelight), thus we modified the Fig.4c to show Chls in a typical Lhcr1 antenna (FCPI-7).

c) cannot present all 24 FCPs since deviations shown in a) are much stronger for the protein backbone. In addition, what do the circles highlight?

Author's answer:

We agree with the reviewer that the deviations of protein backbones are much stronger among the 24 FCPs. Nevertheless, we superposed them in Fig. 4a to show the image of these deviations. We indicated the meaning of the circles in the legend to Fig. 4f.

d) and e) what are green and blue Chls, respectively? What do the circles highlight? Why are Chl 408/401 coloured differently in d) versus e/f?

Author's answer:

Green and blue Chls are Chls *a* and Chls *c*, respectively. The blue cycle in Fig. 4d indicate Chl 416 site that can coordinate both Chl *a* and Chl *c* (extended Table 3). Different color of Chl 408/401 also indicate these two sites can coordinate both Chl *a* and Chl *c* (extended Table 3). We added these into the figure legends.

g) What do the circles highlight?

Author's answer:

We modified the pigments in Fig.4f to avoid confusion, and now we show the location of Cars in a typical Lhcr1 antenna (FCPI-7). In addition, Fx306/307 sites typical in Lhcf-type antennae (Wang et al 2019; Pi et al. 2019) are highlighted by circles.

Figure 5 c and d: I guess it should read: "Labels depict Chls involved in EET between subunits", since all other Chl should work in EET as well

Author's answer:

We modified this sentence as your suggested; thank you.

Extended Material

Figure 1: b) please provide wavelength at which pigments were detected in HPLC,
c) on figure: thylakoid (not thylokoid)

Author's answer:

We detected the pigments at 445 nm. This was added into the revised manuscript (lines 428-429). The typo has been corrected on Extended Figure 1d.

Figure 3b: Lhcr1 instead of Lhca1?

Author's answer:

We added more explanations in the legend of extended Fig. 3b. We compared the structural differences between individual FCPs with Lhcr proteins from a red alga *C. merolae*, Lhca protein from pea and lhcf from *P. tricornutum*. The Lhca1 in extended Fig. 3b was correct.

Table 2: please format in a way that the first row is legible

Author's answer:

The reviewer may indicate the first row of Table 3, and we formatted it to make it legible.

Responses to the comments of reviewer #2

Photosynthesis is a unique energy conversion process performed by plants, algae, and cyanobacteria where photons from the sunlight are converted into chemically fixed energy with a generation of oxygen as a side-product of water splitting. Diatoms being microalgae are widely spread all over the Earth, including water and soil habitats, produce annually up to half of all planet oxygen and ocean organic compounds.

Thus, the organism in the reviewed research is of interest to a wide circle of scientific society. The article revealed the structure of the photosystem I of diatoms.

The photosystem I (PSI) is a large multi-subunit membrane-embedded complex playing the key role in the photosynthetic process. It carries out the light conversion steps that ultimately lead to the production of ATP by ATP-synthase and reduction of NADP to NADPH. In the reviewed article the PSI-FCPI supercomplex of diatoms was resolved at an extremely-high resolution about 2.38 Å, revealing the structural details of a PSI and surrounding FCPI units (PSI-FCPI supercomplex).

It appeared that PSI-FCPI supercomplex is very big and possesses similarities in structural organizations from both red algae and cyanobacteria. At the time, some unique structural differences, common for diatom, namely for *Chaetoceros gracilis*, in both PSI and FCPI organization, determine its physiology, ecology, and evolution. These differences include the presence or absence of some PSI periphery subunits, which are mainly found at higher plants, cyanobacteria, and red algae, the unique organization of FCPI units surrounding PSI, etc.

The special attention in this article is paid to the FCPI subunits. They are very numbered (24 in total) what makes the PSI-FCPI supercomplex structure exceptional in terms of size and complicity.

Recently the PSII-FCPII supercomplex structure of *C. gracilis* was revealed at 3.3 Å resolution (Nagao et al., 2019). The solved structure shows a huge pigment network, and extensive protein-protein, pigment-protein, and pigment-pigment interactions within the PSII-FCPII supercomplex.

In the case of PSI-FCPI supercomplex, the FCPI subunits cannot form higher-order structures (e.g., homotetramer) as it is described in PSII-FCPII (Nagao et al., 2019). At the same time, FCPI subunits in PSI-FCPI supercomplex have original layer organization, surrounding PSI.

There are apparent differences in FCP organization in PSI-FCPI, and PSII-FCPII supercomplexes reflect their different functions during electron transport.

The FCPI subunits in PSI-FCPI supercomplex are monomeric and have original layer organization, around PSI supercomplex.

Thus, this article will bring an inside into the understanding functioning of diatoms electron transport chain.

All in all, I consider the reviewed article is of importance to the scientific community in particular structural biologist and plant physiologist and would assess the work as recommended for the publication in the Nature Communications journal after the

appropriate corrections.

At the same time, I would like to point out that:

1. Chl *a/c* and Fx can absorb light not only in the blue-green part of the visible spectrum (as stated here 67-68) but also in yellow-green one for Fx.

Author's answer:

First of all, we appreciate the reviewer's highly positive and encouraging comments on our manuscript. We agree with your comments, and modified the original sentences as follows (lines 65-68, revised manuscript):

"Chl *a/c* and Fx has rather strong absorptions in the region of 400-550 nm, enabling these organisms survival efficiently under aquatic environments where red light is diminished and the light in the region of 400-550 nm is more available."

2. The lack of structure/function information of the novel PsaS subunit (130). As we see from the main Fig. 1c and Extended data, Fig. 2b, the PsaS subunit is one of the lowest resolved. The reason can be the flexibility of this subunit, the radiation damage, issues connected to image processing steps. PsaS subunit is being in the center of PSI-FCPI supercomplex, located near to the core PsaB subunit in the text of the article named as having the 'peripheral location' (132). What does that mean?

Author's answer:

We describe PsaS as the 'peripheral location' because of its extrinsic location relative to the PSI core and Psa C/D/E subunits. In Figure 1c, we can see that PsaS is located in an outermost position of the core.

3. The representation of the Fig. 1 | overall structure of the PSI-FCPI supercomplex (497) – file name 251291_0_art_file_3393120_q7fkfr.pdf.

In that figure, we see the structure of PSI-FCPI from 1a – stromal, 1b - luminal side. However, on the 1a only the peripheral FCPI are named, whereas the subunits of PSI are only colored and have no name on them. All the PSI-FCPI supercomplex represented as sticks and bounds.

On the 1b the representation is different - FCPIs are numbered, colored in 3 different colors, which probably reflect inner-most / second layer / outer layers of FCPI (no explanation on Fig 1) and shown as a ribbon. The PSI subunits are shown as one grey surface and named. Since we cannot see all the subunits equally (e.g. PsaS) from both stromal and luminal sides, the stromal and luminal sides might be represented consistently, in the same style (e.g., in 1b have a PSI surface separated on subunits with the same color as 1a, etc.). If it is not possible please (due to the space for example) have 2 representations for each side.

Author's answer:

Thank you for your advices. We changed Figure1b according to your suggestion. We marked all the subunits in Fig. 1a and Fig. 1b. A bottom view of Fig.1a, has been

added to make it easy to read. In addition, we added Fig.1d to show the three color annotations of FCPIs.

4. The separation FCPI on the 3 layers (147) (based on their proximity to the PSI? Not explained) and V groups based on the structures and locations (171) is somewhat confusing.

First, the layers are named as:

- inner-most (148): FCPI-1-11;
- the second layer (149) (sometimes named later as middle, why not when the first time introduced?): FCPI-11(again? I guess should be FCPI-12)-22;
- outer-most (150) / but sometimes out-most (278): FCPI-22-24;

-Second, when there is a description of group III FCPI-14/16 (196) the FCPI-14 connects with FCPI-22 from the outer layer and FCPI-13 that named as being from the inner layer (198). I guess there is a misleading, also in Fig. 1b (497-498).

Author's answer:

The separation of FCPI into three layers was based on their proximity to the PSI as you suggested. We added a sentence to indicate this (lines 152-153 in the revised manuscript). We standardized the naming of the three layers. They are innermost layer (FCPI-1 to FCPI-11), middle layer (FCPI-12 to FCPI-21), and outermost layer (FCPI-22-FCPI-24), throughout the text. We also marked them in Fig. 1.

Yes, we made a mistake to name FCPI-13 as being from the inner layer, and we deleted it in the revised manuscript (line 207 in the revised manuscript).

5. PSI-FCPI purification procedure: for the thylakoid solubilization the 0.02% β -DDM was used. Is there substantial evidence that the resulting PSI-FCPI supercomplex contains all the FCPI subunits? You could also try the final solubilization with different β -DDM detergent concentrations in order to see whether PSI-FCPI supercomplex that you get is the only one functional and as close to the native conditions as possible.

Author's answer:

We used 1.0% β -DDM to solubilize thylakoid membranes. The 0.02% β -DDM was used in the solutions of sucrose density gradient centrifugation and subsequent gel filtration. There was no substantial evidence that the resulting PSI-FCPI supercomplex contains all the FCPI subunits, but we consider that our PSI-FCPI represents its native form from the fact that little free Chl and Fx pigments were observed in the sucrose density centrifugation.

6. Did you perform any negatively stained screening (recommended for fast sample evaluation)? Was PSI-FCPI supercomplex the only one present in the cryo-sample under the used conditions?

Author's answer:

Yes, we performed negatively stained screenings many times for sample evaluation. We found that the PSI-FCPI supercomplex we observed represents, if not only, majority of the particles. The following is a typical negatively stained image, which shows that most of particles are the side view and top view of biggest PSI-FCPI supercomplex. Some particles with little smaller sizes may represent other intermediate views of the biggest PSI-FCPI supercomplex. Only a few smaller particles may represent smaller PSI-FCPI supercomplexes, which are removed during the procedure of data processing.

7. The PSI-FCPI purification procedure worked at 4°C or on ice (584). What is the certain reason to freeze the sample at 8 C°?

Author's answer:

We did not freeze the sample at 8°C. This is the temperature at which the Vitrobot Mark IV was kept to work; we modified the manuscript accordingly (line 451 of revised manuscript).

8. 251291_0_extended_data_3393121_q7fkfr.pdf

Extended data Fig. 2b. The local resolution PSI-FCPI supercomplex 3D map looks

quite solid. As expected, most of the FCPIs peripheral subunits are lower resolved. Nonetheless, how can be explained the fact that some of the FCPIs peripheral subunits, all arising from the inner-most layer, namely FCPI 9;10;11 have higher resolution, comparable with the central part of the PSI-FCPI supercomplex? Can it be that those FCPI subunits are not edge-localized under the natural conditions? Can the reason be in the wrong angular assignment of single particles, sample heterogeneity, flexibility, or radiation damage?

You can answer some of those questions, processing the 3D map in Monores and MonoDir.

Author's answer:

It is true that while most of the peripheral FCPIs subunits have lower resolution than the central part of the PSI-FCPI supercomplex, FCPI 9;10;11 have higher resolution comparable with the central part of the PSI-FCPI supercomplex. We consider that this is due to their locations in the innermost layer, but not related with the wrong angular assignment of single particles and sample heterogeneity, etc. for the peripheral FCPI subunits.

9. Extended data Fig. 2b. –the local resolution 3D map is shown for the luminal side only. Please add the stromal side illustration. Also, add the description accordingly.

Author's answer:

The local resolution 3D map shown for the stromal side illustration was added as Extended data Fig. 2d, and the descriptions are added in the legends.

10. Extended Data Fig. 2c – Which FSC do we observe? Please add the FSC plot the will have FSC corrected, unmasked, masked (you can get this information from relion 3.0 that you used for the postprocessing).

Author's answer:

We added FSC plot of corrected, unmasked and masked into Extended Fig. 2c.

11. Extended Data Fig. 2c – Why do FSC plot has a small gap in the low range of frequencies? What determines that?

Author's answer:

We agree with the reviewer that the FSC plot has a small gap in the low range of frequencies because of using a little bit tight mask during Post-processing. If a softer mask is used, the FSC plot may have a smaller gap, but because the gap is very small, we did not change it.

12. Why wasn't the Bayesian polishing step performed after the particle CTF parameters refinement and correction of estimated beam tilt?

Author's answer:

After the particle CTF parameters refinement and correction of estimated beam tilt, the Bayesian polishing step was performed, but the resolution was not improved, so we did not mention it in the original Methods.

13. Please add angular distribution plot for the final reconstruction, so that we could estimate angular coverage of the projection sphere.

Author's answer:

We added the angular distribution plot for the final reconstruction in Extended Data Fig. 2f.

REVIEWER COMMENTS

Reviewer #1 (Remarks to the Author):

The revised manuscript by Xu et al takes into account most points that were raised. However, unfortunately some issues remained:

Meanwhile, the manuscript also from Prof Shen and colleagues that accompanied the first version has been published (Nagao et al 2020). The inner part of the complexes, PSI core and the inner FCPs, are the same – as is to be expected having used the same diatom species. Nonetheless there are differences in the cofactors that were modelled in case of the inner FCPs. In contrast to what is written in answering my first remarks, this issue concerns not only the re-attribution of Chl c to Chl a or vice versa and the detection of more carotenoids. I accept that for carotenoids the differences might be due to the difficulties at this resolution – 2.4 Å for the Nagao structure, overall 2.38 Å for the structure described here, but with higher resolution (up to 2.2) in the inner parts. But it is hard to imagine why the numbers of porphyrins, i.e. chlorophylls, are differing in both directions (Fcp-I-6 = Fcpa6 (Nagao): 13 versus 14, Fcp-I-4 = Fcpa8: 15 vs 14, Fcp-I-14 = Fcpa11: 10 vs 9). Since the whole interpretation of the EET network is based on the number, localisation and assignment of the different Chls, this point is very important and authors should explain this better, comment on the differences, and, if needed insert a sentence of caution concerning those assignments. To say it more simple: there are many readers who do not know much about structural model building and about which resolution allows which statements, but use the models for their research. They need an answer to the question which numbers they should believe and why, and which assignment might be less clear.

Not only the Nagao et al paper is published, but also the sequences retrieved from *C. gracilis* have been submitted to NCBI and can be found there. However, names differ extremely (e.g. FCPI-11 is given here as CgLhcr3, whereas the FCP found in the same location in Nagao's paper, called Fcpa1, is encoded by a gene called Lhcr1). Since the two authoring groups are related, it would be of great benefit for the scientific community if they could use the same nomenclature. Since the pdb file is not available yet, I cannot judge if really the same protein sequences were used for the same subunits. If so, the gene/protein names should be adjusted. If not, an explanation is needed why different sequences were used for modelling.

The second point which was not solved is the tree shown in the supplementary data, which is still not up to standard. The answer "We have used all diatom Lhc family members queried from NCBI and Uniprot databases in the tree (excluding repetitive sequences) to build the tree" is simply not true – for example sequences from *Thalassiosira pseudonana* are mostly missing (except those which were used for structural modelling). And they are not at all repetitive with *P. tricornutum* sequences, even if the names might suggest this. The Lhc of centric diatoms like *T. pseudonana* and *C. gracilis* group differently to those of pennate diatoms like *P. tricornutum*, as shown by many papers. Thus *Thalassiosira* cannot be omitted in such a phylogenetic analysis (and in cases where sequences of *C. gracilis* and *T. pseudonana* are identical, they could be labelled as such). This might also solve the "predicted" FCP issue, since FCP2 etc are the proper names given in case of a family of *T. pseudonana* Lhcs.

Reviewer #2 (Remarks to the Author):

The author team has revised and corrected most of the critical remarks that I made in a my previous review.

I have no further comments to add.

I consider this manuscript as prepared for further publication.

With kind regards,
Dr. Dmitry Semchonok

Responses to the comments of Reviewer #1

Meanwhile, the manuscript also from Prof Shen and colleagues that accompanied the first version has been published (Nagao et al 2020). The inner part of the complexes, PSI core and the inner FCPs, are the same – as is to be expected having used the same diatom species. Nonetheless there are differences in the cofactors that were modelled in case of the inner FCPs. In contrast to what is written in answering my first remarks, this issue concerns not only the re-attribution of Chl c to Chl a or vice versa and the detection of more carotenoids. I accept that for carotenoids the differences might be due to the difficulties at this resolution – 2.4 Å for the Nagao structure, overall 2.38 Å for the structure described here, but with higher resolution (up to 2.2) in the inner parts. But it is hard to imagine why the numbers of porphyrins, i.e. chlorophylls, are differing in both directions (Fcp-I-6 = Fcpa6 (Nagao): 13 versus 14, Fcp-I-4 = Fcpa8: 15 vs 14, Fcp-I-14 = Fcpa11: 10 vs 9).

Since the whole interpretation of the EET network is based on the number, localisation and assignment of the different Chls, this point is very important and authors should explain this better, comment on the differences, and, if needed insert a sentence of caution concerning those assignments. To say it more simple: there are many readers who do not know much about structural model building and about which resolution allows which statements, but use the models for their research. They need an answer to the question which numbers they should believe and why, and which assignment might be less clear.

Not only the Nagao et al paper is published, but also the sequences retrieved from *C. gracilis* have been submitted to NCBI and can be found there. However, names differ extremely (e.g. FCPI-11 is given here as CgLhcr3, whereas the FCP found in the same location in Nagao's paper, called Fcpa1, is encoded by a gene called Lhcr1). Since the two authoring groups are related, it would be of great benefit for the scientific community if they could use the same nomenclature. Since the pdb file is not available yet, I cannot judge if really the same protein sequences were used for the same subunits. If so, the gene/protein names should be adjusted. If not, an explanation is needed why different sequences were used for modelling.

Author Ans:

We thank the reviewer for pointing out this, and explained the differences in the revised manuscript in more detail. We added supplementary Table 4 and supplementary Fig. 7 to compare the sequences of FCPIs used in the present study and those of Nagao et al., and the assignment of Chls in the two structures. The more Chls assigned in FCPI-11 (two more), FCPI-14, FCPI-16 (one each) than the corresponding FCPI-1, FCPI-11, FCPI-13 of Nagao et al.'s structure are located at the periphery region, and will not affect the energy transfer pathways (Supplementary Fig. 7). The more Chls

assigned in FCPI-6, FCPI-7 (one each) in Nagao et al.'s structure than the corresponding FCPI-6, FCPI-5 in the current structure are located within the bulk Chls and could affect the energy transfer pathways; however, they may be mis-assigned in the previous structure based on the low resolution map. The Chls in FCPI-4 of the current structure was the same as that in the corresponding FCPI-8 of Nagao et al.'s structure, but it was mis-listed as one Chl less in the previous structure. We added a section to the revised manuscript to explain these (last paragraph, page 11-12).

The sequences of FCPIs used in the present study are mostly the same as those used in the Nagao et al.'s study, with only a few amino acid residues different (Supplementary Table 4). For example, the sequences of FCPI-11 used in the present study was the same as the Fcpa1 sequences used in the Nagao et al.'s study, except 3 amino acid residues. This difference may be due to incorrect sequences resulted from the present cDNA sequencing, or sequencing errors in the previous study.

The second point which was not solved is the tree shown in the supplementary data, which is still not up to standard. The answer "We have used all diatom lhc family members queried from NCBI and Uniprot databases in the tree (excluding repetitive sequences) to build the tree" is simply not true – for example sequences from *Thalassiosira pseudonana* are mostly missing (except those which were used for structural modelling). And they are not at all repetitive with *P. tricornutum* sequences, even if the names might suggest this. The Lhc of centric diatoms like *T. pseudonana* and *C. gracilis* group differently to those of pennate diatoms like *P. tricornutum*, as shown by many papers. Thus *Thalassiosira* cannot be omitted in such a phylogenetic analysis (and in cases where sequences of *C. gracilis* and *T. pseudonana* are identical, they could be labelled as such). This might also solve the "predicted" FCP issue, since FCP2 etc are the proper names given in case of a family of *T. pseudonana* Lhcs.

Author Ans:

Based on the reviewer's comments, we added sequences from *Thalassiosira pseudonana* and some other species into the tree. The results showed no changes in the grouping of FCPIs. We replaced the tree by a new one (Supplementary Fig. 5b).